# CIM: Constrained Intrinsic Motivation for Reinforcement Learning

## Abstract

This paper investigates two fundamental problems that arise when implementing intrinsic motivation for reinforcement learning: 1) how to design a proper intrinsic objective in Reward-Free Pre-Training (RFPT) tasks, and 2) how to reduce the bias introduced by the intrinsic objective in Exploration with Intrinsic Motivation (EIM) tasks. Existing intrinsic motivation methods suffer from static skills, limited state coverage, sample inefficiency in RFPT, and suboptimality in EIM. To tackle these problems, we propose *Constrained Intrinsic Motivation (CIM)* for RFPT and EIM, separately. CIM for RFPT maximizes a novel lower bound of the conditional state entropy with a new alignment constraint on the skill and state representations for efficient dynamic skill discovery and state coverage maximization. CIM for EIM leverages constrained policy optimization to adaptively adjust the temperature parameter of the intrinsic reward for bias reduction. In multiple MuJoCo robotics environments and tasks, we empirically show that CIM for RFPT achieves greatly improved performance and sample efficiency over state-of-the-art intrinsic motivation methods. Additionally, we showcase the effectiveness of CIM for EIM in redeeming intrinsic rewards when extrinsic rewards are exposed from the beginning.

## 1 Introduction

In the realm of Reinforcement Learning (RL), Intrinsic Motivation (IM) assumes a vital role in the design of exploration strategies (Barto, 2013). IM operates by formulating the agent's familiarity with the environment as the intrinsic objective and employing the intrinsic bonus as a measure of uncertainty for curiosity-driven exploration. It allows agents to efficiently visit novel regions by assigning higher bonuses to unfamiliar states in a principled way (Liu & Abbeel, 2021b; Zhang et al., 2021). Practical IM methods can be classified into knowledge-based, data-based, and competence-based (Laskin et al., 2021).

Knowledge-based IM methods approximate the novelty of a state, analogous to the principled UCB bonus, by maximizing deviation from explored regions (named as policy coverage) $\rho_\pi^{-1}$ (Zhang et al., 2021). Common approximation approaches include the pseudo-count of state visit frequency (Bellemare et al., 2016; Fu et al., 2017), prediction errors of specific neural networks such as ICM (Pathak et al., 2017) or RND (Burda et al., 2018), and variances of outputs within an ensemble of neural networks (Pathak et al., 2019; Lee et al., 2021; Bai et al., 2021). However, existing knowledge-based IM methods may encounter issues such as detachment, derailment (Ecoffet et al., 2021), and catastrophic forgetting (Zhang et al., 2021). Moreover, knowledge-based IM methods are inefficient in Reward-Free Pre-Training (RFPT) task due to their lack of awareness of latent skill variables (Laskin et al., 2021).

Data-based IM methods, on the other hand, directly incentivize the agent to achieve high state coverage by maximizing the state entropy $H(s)$ (Hazan et al., 2019; Mutti et al., 2021; Liu & Abbeel, 2021a;b; Seo et al., 2021). However, these methods also do not condition latent skill variables in RFPT tasks, limiting the applicability of the pre-trained policy for downstream tasks. Additionally, exploration with knowledge- or data-based IM methods introduces non-negligible biases that can lead to suboptimal policies in Exploration with IM (EIM) tasks. Specifically, intrinsic objectives can lead to superfluous exploration even when the task rewards are already accessible. This distrac-

tion, induced by intrinsic objectives, can deteriorate the performance of the agent. Consequently, it can impede the wider application of knowldege- and data-based IM methods.

Competence-based IM methods primarily maximize Mutual Information (MI) $I(\mathrm{s};\mathrm{z})$ between the state $s$ (or the trajectory $\tau$) and the latent skill variable $z$. A policy conditioned on latent skill variables can change the state of the environment in a consistent and meaningful way, e.g., walking, jumping, flipping, pushing., and thus can be efficiently finetuned to solve downstream tasks. However, these methods have shown poor performance in the Unsupervised Reinforcement Learning Benchmark (URLB) (Laskin et al., 2021). Intuitively, directly maximizing MI does not guarantee extensive state coverage or the discovery of dynamic skills, as evidenced by recent unsupervised skill discovery researches (Laskin et al., 2021; Park et al., 2021; 2023). Due to the invariance of MI to scaling and invertible transformation of the input variables, maximizing only MI will easily converge to simple and static skills. To address this limitation, former works like DIAYN (Eysenbach et al., 2018) and DADS (Sharma et al., 2019) utilize an inductive bias known as the $x - y$ prior to make the agent focus only on $x - y$ related primitives. Park et al. (2021) proposed LSD based on Lipschitz-constrained state representation learning to avoid the usage of the $x - y$ prior. However, LSD suffers from severe sample inefficiency even when using the off-policy RL method.

To overcome the limitations mentioned above, in this paper, we propose *Constrained Intrinsic Motivation (CIM)* which 1) constructs a proper constrained intrinsic objective via a lower bound of the state entropy to allow the agent to be aware of the latent skill variable while maximizing the state entropy, and 2) adaptively balancing the intrinsic and extrinsic objectives according to the performance of the agent when the task rewards are exposed in the beginning.

In summary, we make the following main contributions:

- We propose *Constrained Intrinsic Motivation (CIM)* to overcome the limitations of data-/knowledge- and competence-based intrinsic motivation by combining the best of both worlds. CIM outperforms state-of-the-art intrinsic motivation methods, improving performance and sample efficiency in multiple MuJoCo robotics environments.

- CIM for RFPT introduces a lower bound for the state entropy that conditions the state entropy on the latent skill variable without compromising the power of maximum state entropy exploration. CIM for RFPT also introduces a novel alignment loss to make dynamic skills interpretable. Compared with LSD (Park et al., 2021) (one state-of-the-art skill discovery method), our CIM reduces the number of required samples from 400M to 20M in the environment Ant and facilitates the running time from $\sim$15 hours to $\sim$10 mins (**90x faster**) with our implementation in the same device. Besides skill diversity and state coverage, our CIM achieves the highest fine-tuning score in the Walker domain of URLB (Laskin et al., 2021).

- CIM for EIM derives an adaptive schedule for the temperature weight of intrinsic rewards leveraging the constrained policy optimization method. We empirically demonstrate that the adaptive schedule can effectively diminish the bias introduced by intrinsic bonuses in various MuJoCo tasks.

## 2 PRELIMINARIES

**Markov Decision Processes.** The discounted Markov Decision Process (MDP) is defined as $M = (\mathcal{S}, \mathcal{A}, P, R, \gamma, \mu)$, where $\mathcal{S}$ and $\mathcal{A}$ stand for the state space and the action space separately, $P : \mathcal{S} \times \mathcal{A} \to \Delta(\mathcal{S})$ is the transition function mapping the state $s$ and the action $a$ to the distribution $P(s'|s, a)$ in the space of probability distribution $\Delta(\mathcal{S})$ over $S$, $R : \mathcal{S} \times \mathcal{A} \times \mathcal{S} \to \mathbb{R}$ is the reward function, $\gamma \in [0, 1)$ is the discount factor, and $\mu \in \Delta(\mathcal{S})$ is the initial state distribution. We focus on the episodic setting where the environment is reset once the agent reaches a final state $s_f$, a terminated state within the goal subsets $\mathbb{G}$ or a truncated state $s_T$. At the beginning of each episode, the agent samples a random initial state $s_0 \sim \mu$; at each time $t = 0, 1, 2, ..., T - 1$, it takes an action $a_t \in \mathcal{A}$ computed by a stochastic policy $\pi : \mathcal{S} \to \Delta(\mathcal{A})$ or a deterministic one $\pi : \mathcal{S} \to \mathcal{A}$ according to the current state $s_t$ and steps into the next state $s_{t+1} \sim P(\cdot|s_t, a_t)$ with an instant reward signal $r_t = R(s_t, a_t, s_{t+1})$ obtained.

**Intrinsically Motivated RL.** Intrinsically Motivated RL (IMRL) aims to maximize the following general objective

$$L_k(\pi) = (1 - \lambda_k)J_e(d_\pi) + \lambda_k J_i(d_\pi), \qquad (1)$$

where $d^\pi(s) := (1 - \gamma) \sum_{t=0}^{\infty} \gamma^t P(s_t = s | \mu, \pi)$ is the state distribution induced by the policy $\pi$, $\mathcal{K}$ is the collection of all induced distributions, $J_e(d_\pi) := \mathbb{E}_{s \sim d_\pi}[r_e]$ is the extrinsic objective defined as the expectation of the extrinsic reward $r_e := R_e(s, a, s')$ over the induced state distribution $d_\pi$, $J_i : \mathcal{K} \to \mathbb{R}$ is the intrinsic objective defined as a differentiable function of the induced state distribution $d_\pi$ with $L-$Lipschitz gradients, and $\lambda_k$ is a coefficient to balance the two objectives. We summarize the choices of the intrinsic objective $J_i$ in current IMRL algorithms in Table 1. The projection network $\phi : \mathcal{S} \to \mathcal{Z}$ utilized in competence-based intrinsic motivation methods is independent of the choice of the state encoder network $f : \mathcal{S} \to \mathcal{H}$ used in state representation learning. For instance, for state-based RL tasks, $s$ represents the raw state vector, $h := f(s) \in \mathbb{R}^m$ represents the state representation vector, and $\phi(s) \in \mathbb{R}^n$ stands for the state projection vector. Note that for competence-based IM methods APS (Liu & Abbeel, 2021a), CIC (Laskin et al., 2022), and MOSS (Zhao et al., 2022), we use $H(\phi(s))$ for the state entropy and $H(\phi(s)|z)$ for the conditional state entropy since they are typically estimated in the state projection space $\mathcal{Z}$ instead of the original state space.

**Reward-Free Pre-Training and Exploration with IM** RFPT (a.k.a. unsupervised reinforcement learning) and EIM are two main branches of IMRL. The objective of RFPT is to pre-train an agent without any task reward $r^e$ available, which can be regarded as IMRL with $\lambda_k \equiv 1$, that is,

$$L_k^{\text{RFPT}}(\pi) = J_i(d_\pi). \qquad (2)$$

In RFPT, the agent aims to learn either a policy $\pi(a|s)$ that maximizes a knowledge- or data-based intrinsic objective or a latent-conditioned policy $\pi(a|s, z)$ that maximizes a competence-based intrinsic objective. Evaluation metrics for RFPT can be state coverage, skill diversity, and fine-tuning performance in downstream tasks. In contrast, EIM refers to training an agent with task rewards available from the beginning, which can be seen as IMRL with $\lambda_k < 1$ for all $k$, that is,

$$L_k^{\text{EIM}}(\pi) = J_e(d_\pi) + \tau_k J_i(d_\pi), \qquad (3)$$

where $\tau_k := \lambda_k/(1 - \lambda_k)$ is the temparature parameter. The evaluation metric for Exploration with IM is the extrinsic objective $J_e(d_\pi)$. Hence, in EIM tasks, the agent commonly uses knowledge- or data-based IM instead of competence-based IM for exploration.

## 3 CONSTRAINED INTRINSIC MOTIVATION

In this section, we design CIM for RFPT and EIM tasks separately. First, we propose a constrained intrinsic motivation $J_i^{\text{CIM}}$ for RFPT to maximize the conditional state entropy with an alignment constraint between the state representation $\phi(s)$ and the latent skill variable $z$. This encourages the agent to learn dynamic skills. We derive the corresponding intrinsic reward $r_i^{\text{CIM}}$ based on the Frank-Wolfe algorithm. Second, we propose constraining IM via the extrinsic objective in EIM and derive an automatic temperature schedule $\tau_k^{\text{CIM}}$ in Equation (3) based on the Lagrangian duality theory.

### 3.1 CONSTRAINED INTRINSIC MOTIVATION FOR REWARD-FREE PRE-TRAINING

In this section, we develop CIM for RFPT. To design the intrinsic objective, we first review current coverage- and MI-based methods and analyze their limitations.

**Limitation of Coverage-Based IM.** We denote knowledge- and data-based IM methods as coverage-based methods since they either maximize deviation from policy coverage or directly maximize state coverage. Though current coverage-based IM methods like RND (Burda et al., 2018) and APT (Liu & Abbeel, 2021b) perform well in terms of state coverage in certain types of environments, these methods lack awareness of latent skill variables and suffer from limited fine-tuning efficiency.

**Limitation of MI-Based IM.** MI-based IM methods distill the agent's exploration experience into useful skills. There are two types of decomposition for MI $I(s; z)$, that is, $I(s; z) = H(z) - H(z|s) = H(s) - H(s|z)$. Methods like DIAYN (Eysenbach et al., 2018) and VISR (Hansen et al., 2019)

Table 1: A summarization of intrinsic motivation algorithms, including: 1) *knowledge-based* intrinsic motivation methods: ICM (Pathak et al., 2017), RND (Burda et al., 2018), Disagreement (Pathak et al., 2019), MADE (Zhang et al., 2021), and AGAC (Flet-Berliac et al., 2021); 2) *data-based* intrinsic motivation methods: MaxEnt (Hazan et al., 2019), APT (Liu & Abbeel, 2021b), and RE3 (Seo et al., 2021); 3) *competence-based* intrinsic motivation methods: VIC (Gregor et al., 2016), DIAYN (Eysenbach et al., 2018), VISR (Hansen et al., 2019), DADS (Sharma et al., 2019), APS (Liu & Abbeel, 2021a), CIC (Laskin et al., 2022), MOSS (Zhao et al., 2022), BeCL (Yang et al., 2023), LSD (Park et al., 2021), CSD (Park et al., 2023), and CIM. $s_T$ represents the last state in one trajectory, where $T$ denotes the final time step in one episode. $s' \sim P(s'|s, a)$ is the subsequent state transitioned from the current state $s$ when action $a$ is taken, and $z$ is the latent skill. In the Intrinsic Objective column, $d_\pi$ stands for the induced state distribution, $\rho$ the policy cover, $D_{\mathrm{KL}}$ the KL-divergence, $H$ the entropy, $f : \mathcal{S} \to \mathcal{H}$ the state encoder network, $\phi : \mathcal{S} \to \mathcal{Z}$ the projection network. In the Intrinsic Reward column, $\hat{d}$ the estimated state distribution, $\hat{\rho}$ the estimated policy cover, $q$ the discriminator, $S_c$ the cosine similarity.

| Algorithm | Intrinsic Objective | Intrinsic Reward |
|---|---|---|
| ICM | $\mathbb{E}_s[\rho_\pi^{-1}(s)]$ | $\hat{\rho}_\pi^{-1}(s)$ |
| RND | $\mathbb{E}_s[\rho_\pi^{-1}(s)]$ | $\hat{\rho}_\pi^{-1}(s)$ |
| Dis. | $\mathbb{E}_s[\rho_\pi^{-1}(s)]$ | $\hat{\rho}_\pi^{-1}(s)$ |
| MADE | $\mathbb{E}_s[(\rho_\pi^{-1}(s)d_\pi^{-1}(s))^{1/2}]$ | $(\hat{\rho}_\pi^{-1}(s)\hat{d}_\pi^{-1}(s))^{1/2}$ |
| AGAC | $\mathbb{E}_s[D_{\mathrm{KL}}(\pi(s)|\pi^a(s))]$ | $D_{\mathrm{KL}}(\pi(s)|\pi^a(s))$ |
| MaxEnt | $H(\mathrm{s})$ | $-\log \hat{d}_\pi(s)$ |
| APT | $H(\mathrm{s})$ | $-\log \hat{d}_\pi(f(s))$ |
| RE3 | $H(\mathrm{s})$ | $-\log \hat{d}_\pi(f(s))$ |
| VIC | $H(\mathrm{z}) - H(\mathrm{z}|\mathrm{s}_T)$ | $\log q(z|s_T)$ |
| DIAYN | $H(\mathrm{z}) - H(\mathrm{z}|\mathrm{s}) + H(\mathrm{a}|\mathrm{s}, \mathrm{z})$ | $\log q(z|s)$ |
| VISR | $H(\mathrm{z}) - H(\mathrm{z}|\mathrm{s})$ | $S_c(\phi(s), z)$ |
| DADS | $H(\mathrm{s}'|\mathrm{s}) - H(\mathrm{s}'|\mathrm{s}, \mathrm{z})$ | $-\log \hat{q}(s'|s) + \log q(s'|s, z)$ |
| APS | $H(\phi(\mathrm{s})) - H(\phi(\mathrm{s})|\mathrm{z})$ | $-\log \hat{d}_\pi(\phi(s)) + S_c(\phi(s), z)$ |
| CIC | $H(\phi(\mathrm{s})),$ s.t. $\phi \in \arg\min L^{\mathrm{CIC}}(\phi(s), z)$ | $-\log \hat{d}_\pi(\phi(s))$ |
| MOSS | $\mathbb{E}_{m \sim \mathcal{B}}(1 - 2m)H(\phi(\mathrm{s})|m)$ | $-(1 - 2m)\log \hat{d}_\pi(\phi(s))$ |
| BeCL | $I(\mathrm{s}; \mathrm{s}^+),$ s.t. $\phi \in \arg\min L^{\mathrm{BeCL}}$ | $\exp(-l^{\mathrm{BeCL}})$ |
| LSD | $\mathbb{E}_{z,s}(\phi(s') - \phi(s))^T z$ | $(\phi(s') - \phi(s))^T z$ |
| CSD | $\mathbb{E}_{z,s}(\phi(s') - \phi(s))^T z,$ s.t. $\phi \in \arg\min L^{\mathrm{CSD}}$ | $(\phi(s') - \phi(s))^T z$ |
| **CIM (ours)** | $H(\phi(\mathrm{s})|\mathrm{z}),$ s.t. $\phi \in \arg\min L^{\mathrm{CIM}}$ | $-\log \hat{d}_\pi(\phi(s)^T z|z)$ |

employ the first type of decomposition. However, they tend to learn static skills like posing or doing yoga, as $H(\mathrm{z})$ is fixed and $H(\mathrm{z}|\mathrm{s})$ can be minimized with slight differences in states. To alleviate this drawback, DADS (Sharma et al., 2019) introduces the $x - y$ prior in the skill dynamic module $q(s'|s, z)$ by inputting only the agent's position. Nonetheless, this approach may neglect other types of skills. On the other hand, LSD (Park et al., 2021) proposes to learn dynamic skills without the $x-y$ prior by maximizing $\mathbb{E}_{z \sim p}\mathbb{E}_{s \sim d_\pi}(\phi(s') - \phi(s))^T z$ to encourage the state differences where $\phi$ is $1-$Lipschitz. However, it suffers from severe sample inefficiency due to the Lipschitz constraint and a lack of explicit maximization of $H(\mathrm{s})$. To address the drawback of the first type of decomposition, APS (Liu & Abbeel, 2021a) turns to the second one, that is, $H(\mathrm{s}) - H(\mathrm{s}|\mathrm{z})$, where $H(\mathrm{s}|\mathrm{z})$ is estimated by modeling $q(\mathrm{z}|\mathrm{s})$ as von-Mises Fisher (vMF) distribution, similar to VISR (Hansen et al., 2019). However, minimizing $H(\mathrm{s}|\mathrm{z})$ impedes the maximization of $H(\mathrm{s})$, discouraging the agent's exploration regarding state coverage. This phenomenon is also found in CIC (Laskin et al., 2022), where choosing $H(\mathrm{s})$ as the intrinsic objective leads to higher fine-tuning efficiency than $H(\mathrm{s}) - H(\mathrm{s}|\mathrm{z})$ in URLB (Laskin et al., 2021). Even when explicitly maximizing $H(\mathrm{s})$, CIC (Laskin et al., 2022) cannot learn dynamic skills compared to LSD (Park et al., 2021).

### 3.1.1 DESIGN OF CONSTRAINED INTRINSIC OBJECTIVE

Motivated by the above analysis and inspired by the fact that $H(s) = H(s|z) + I(s;z)$, we propose maximizing a lower bound of conditional state entropy $H(s|z)$ and a lower bound of MI $I(s;z)$ at the same time. This approach encourages the agent to maximize state coverage and distill dynamic skills simultaneously. To alternately maximize the two lower bounds, we choose the conditional state entropy $H(s|z)$ as the intrinsic objective and utilize an alignment constraint to align the state representation and the latent skill, that is,

$$J_i^{\text{CIM}} = H^\pi(\phi(s)|z) = \mathbb{E}_{z,\phi(s)\sim d_{\pi(\cdot|s,z)}}[-\log d_{\pi(\cdot|s,z)}], \quad \text{s.t.} \quad \phi(s) \in \arg\min L_a(\phi(s), z). \quad (4)$$

where $H^\pi(\phi(s)|z)$ the conditional state entropy estimated in the state projection space $\mathcal{Z}$ which is depends on the policy network $\pi$ and the state projection network $\phi$, $d_{\pi(\cdot|s,z)}$ is the state distribution induced by the latent-conditioned policy $\pi(\cdot|s,z)$, $L_a := \sum_i l_i$ is the alignment loss. This formulation provides a novel insight to unify former MI-based methods. Here, we list all former choices for $l_i$ in competence-based IM methods,

$$
\begin{aligned}
l_i^{\text{MSE}} &= \|\phi(s_i') - z_i\|_2^2 = -\phi(s_i')^T z_i + \|\phi(s_i')\|^2 + \|z_i\|^2, \\
l_i^{\text{vMF}} &= -S_c(\phi(s_i'), z_i), \\
l_i^{\text{LSD}} &= -\phi^{\text{diff}}(\tau_i)^T z_i + \lambda(\|\phi^{\text{diff}}(\tau_i)\| - d(s, s')), \\
l_i^{\text{CIC}} &= -S_c(\phi(\tau_i), \phi_z(z_i)) + \log \sum_{\tau_j \in S^- \bigcup\{\tau_i\}} \exp\Big(S_c(\phi(\tau_j), \phi_z(z_i))\Big), \\
l_i^{\text{BeCL}} &= -S_c(\phi(s_i^+), \phi(s_i)) + \log \sum_{s_j \in S^- \bigcup\{s_i^+\}} \exp\Big(S_c(\phi(s_j), \phi(s_i))\Big),
\end{aligned}
\quad (5)
$$

where $\tau := (s, s')$ is the slice of a trajectory and $\phi^{\text{diff}}(\tau) := \phi(s') - \phi(s)$ is the state-difference version of the trajectory representation, and $d(s, s')$ is the state distance which can be chosen as $\|s' - s\|$ or $-\log(s'|s)$. VIC (Gregor et al., 2016) and DIAYN (Eysenbach et al., 2018) utilize $l_i^{\text{MSE}}$ to train the skill discriminator $q(z|s)$. VISR (Hansen et al., 2019) and APS (Liu & Abbeel, 2021a) use $l_i^{\text{vMF}}$ to learn $q(s|z)$ and $q(z|s)$ separately. LSD (Park et al., 2021) and CSD (Park et al., 2023) adopt $l_i^{\text{LSD}}$ as the alignment loss. CIC (Laskin et al., 2022) and MOSS (Zhao et al., 2022) use $l_i^{\text{CIC}}$ to align the trajectory representation and the projected skill. Lastly, BeCL (Yang et al., 2023) uses $l_i^{\text{BeCL}}$ to align multi-view state representations. To encourage large state difference and maximize MI at the same time, we design a novel alignment loss function as follows

$$l_i^{\text{CIM}} = -\phi^{\text{diff}}(\tau_i)^T z_i + \log \sum_{\tau_j \in S^- \bigcup\{\tau_i\}} \exp\left((\phi^{\text{diff}}(\tau_j))^T z_i\right). \quad (6)$$

This loss function is based on Contrastive Predictive Coding (CPC) by regarding the latent skill $z$ as the context and adopting the state-difference version of trajectory representation $\phi_\tau^{\text{diff}}(s, s')$ as the predictive coding. $S^-$ is a set of negative samples that contains trajectories sampled via skills other than $z_i$. It can be easily proven based on CPC that minimizing the CIM loss function $l_i^{\text{CIM}}$ maximizes a lower bound of MI $I(s;z)$, that is, $I(s;z) \geq \log N - \sum l_i^{\text{CIM}}$, where $N$ is the total number of samples for estimating the MI.

### 3.1.2 ESTIMATION OF CONDITIONAL STATE ENTROPY

We now explain how to estimate the conditional state entropy $H(s|z)$ involved in the intrinsic objective of CIM and then derive the intrinsic bonus of CIM for RFPT tasks.

Recall the definition of the conditional state entropy $H^\pi(\phi(s)|z) = \mathbb{E}_{z\sim p_z}[H^\pi(\phi(s)|z=z)] = \mathbb{E}_{z\sim p_z}\left[\mathbb{E}_{\phi(s)\sim d_{\pi(\cdot|s,z)}}[-\log d_{\pi(\cdot|s,z)}]\right]$. To estimate the outer expectation, we randomly sample the latent skill variables $z$ from a prior distribution $p_z(z)$. For discrete skills, $p_z(z)$ can be a categorical distribution $\text{Cat}(K, \boldsymbol{p})$ that is parameterized by $\boldsymbol{p}$ over a size-$K$ the sample space, where $p_i$ denotes the probability of the $i-$th skill. For continuous skills, we can select $p(z)$ as a uniform distribution $\mathcal{U}^{n_z}(a, b)$ over the interval $[a, b]$, where $n_z$ is the dimension of the skill. To estimate the inner expectation, we roll out trajectories using the policy $\pi(\cdot|s, z)$ with $z$ fixed. To estimate the state density $d_{\pi(\cdot|s,z)}$, instead of training a parameterized generative model, we leverage a more practical non-parametric $\xi-$nearest neighbor ($\xi-$NN) estimator, that is,

$$\hat{d}_{\pi(\cdot|s,z)}(s_i) = \frac{1}{\lambda\left(B_\xi(s_i)\right)} \int_{B_\xi(s_i)} d_{\pi(\cdot|s,z)}(s)\mathrm{d}s \quad (7)$$

where $\lambda$ is the Lebesgue measure on $\mathbb{R}^d$, $B_\xi$ is the smallest ball centered on $s_i$ containing its $\xi$-th nearest neighbour $s_i^\xi$.

**Proposition 1.** *Given a deterministic g, $H(\mathrm{s}|\mathrm{z} = z) \geq H(g(\mathrm{s})|\mathrm{z} = z))$ with equality iff g is invertible.*

**Proposition 2.** *Given a deterministic g, $H(\mathrm{s}|\mathrm{z}) \geq H(g(\mathrm{s})|\mathrm{z}))$ with equality iff g is invertible.*

**Theorem 3.** *Given $g(\phi(s)) := \max(\phi(s)^T z, 0)$, $H^\pi(\phi(\mathrm{s})|\mathrm{z}) \geq H^\pi(g(\phi(\mathrm{s}))|\mathrm{z}))$ with equality iff $\phi(s)^T z = \|\phi(s)\|\|z\|$.*

To encourage the agent to learn dynamic skills, we further define a projection function $g(\phi(s)) := \max(\phi(s)^T z, 0)$ for a fixed skill $z$. Based on Theorem 3, $H(\phi(s)|\mathrm{z})$ is a lower bound of $J_i^{\mathrm{CIM}}$ and the bound is tight when the state representation $\phi(s)$ and the skill $z$ is well aligned (i.e., $\phi(s)^T z = \|\phi(s)\|\|z\|$). The proof of Theorem 3 are provided in Appendix D. The intrinsic reward of CIM for RFPT is then $r_i^{\mathrm{CIM}}(s) = \log \|g(\phi(s)) - g(\phi(s))^\xi\|$. Here, $g(\phi(s))^\xi$ means the $\xi$-th nearest neighbor of $g(\phi(s))$. We adopted an average-distance version similar to APT to make training more stable; that is,

$$r_i^{\mathrm{CIM}}(s) = \log \left( 1 + \frac{1}{\xi} \sum_{j=1}^\xi \|g(\phi(s)) - g(\phi(s))^j\| \right). \tag{8}$$

Intuitively, $r_i^{\mathrm{CIM}}(s)$ measures how sparse the state $s$ is in the positive half-space spaned by its corresponding latent skill $z$. This reward function can be justified based on the procedure of the Frank-Wolfe algorithm when solving Equation (1). Specifically, since $L_k$ is concave in $d_\pi$, maximizing $L_k$ involves solving $d_{\pi_{k+1}} \in \arg \max \langle \nabla_{d_\pi} L(d_{\pi_k}), d_{\pi_k} - d_\pi \rangle$ iteratively (Hazan et al., 2019). This iterative step is equivalent to policy optimization using a reward function proportional to $\nabla_{d_\pi} L(d_{\pi_k})$.

### 3.2 Constrained Intrinsic Motivation for Exploration

In this section, we present our CIM for EIM tasks. In EIM tasks, it is effective to use coverage-based IM methods such as RND and APT to encourage the agent to explore novel states. However, in these methods, the intrinsic reward $\nabla H(s)$ can never converge to zero, which introduces an unvanishing bias to the learned policy. This, in turn, makes the policy suboptimal and requires the temperature parameter $\tau_k$ to be adaptively decreased. Currently, IM methods use a constant temperature parameter or apply a task-specific linear or exponential decay schedule. To avoid the cost of hyperparameter tuning, we propose an adaptive schedule based on the performance of the agent. Specifically, we reformulate Equation (3) by regarding the extrinsic objective as a constraint for the intrinsic objective as follows

$$\max_{d_\pi \in \mathcal{K}} J_i(d_\pi), \text{ s.t. } J_e(d_\pi) \geq R_k \tag{9}$$

where $R_k$ represents the expected reward at the $k$-th step of policy optimization, which can be approximated via $\hat{R}_k = \max_{j \in \{1,2,...,k-1\}} J_e(d_{\pi_j})$. For a comparison between our proposed constraint and Extrinsic Optimality Constraint proposed by Chen et al. (2022), please refer to Appendix G. We then leverage the Lagrangian method to solve Equation (9). The corresponding Lagrangian dual problem is $\min_{\lambda \geq 0} \max_{d_\pi} J_i(d_\pi) + \lambda_k(J_e(d_\pi) - \hat{R}_k)$. The Lagrangian multiplier $\lambda$ is updated by Stochastic Gradient Descent (SGD), that is, $\lambda_k = \lambda_{k-1} - \eta(J_e(d_{\pi_k}) - \hat{R}_{k-1})$. Observing that $\mathcal{L}_k(d_\pi, \lambda_k) \propto J_e(d_\pi) + \lambda_k^{-1} J_i(d_\pi)$, the adaptive temperature $\tau_k^{\mathrm{CIM}}$ is then derived as

$$\tau_k^{\mathrm{CIM}} = \min\{\lambda_k^{-1}, 1\} = \min\{\{\lambda_{k-1} - \eta(J_e(d_{\pi_k}) - \hat{R}_{k-1})\}^{-1}, 1\}, \tag{10}$$

where the outer minimization is to ensure numerical stability. It is worth noting that $\tau_k^{\mathrm{CIM}}$ is the inverse of $\lambda_k$. Thus, as $\lambda_k$ grows, $\tau_k$ gradually tends to zero, that is, the bias introduced by the intrinsic objective $J_i$ in EIM tasks is adaptively reduced.

## 4 Experiments

We design comprehensive experiments to evaluate our competence-based intrinsic method CIM for RFPT and our adaptive temperature scheduler CIM for EIM.

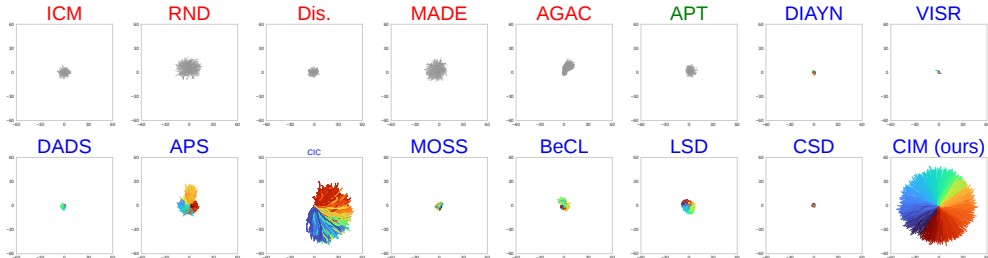

Figure 1: Visualization of 2D continuous locomotion skills discovered by various IM methods in Ant. Each color in competence-based methods represents the direction of the skill latent variable $z$.

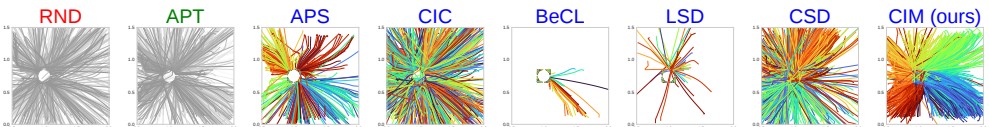

Figure 2: Visualization of 2D continuous manipulation skills discovered by various IM methods in FetchSlide. Each color in competence-based methods represents the direction of the skill $z$.

## 4.1 EXPERIMENTAL SETUP

### 4.1.1 EXPERIMENTAL SETUP FOR RFPT

**Environments.** We evaluate our intrinsic bonus $r_i^{\text{CIM}}$ for RFPT tasks on four Gymnasium environments, including two locomotion environments (Ant and Humanoid) and two manipulation environments (FetchPush and FetchSlide). For more information on all the environments we used, please see Appendix A.

**Baseline Methods and Implementation Details.** We compare CIM for RFPT with fifteen different IM methods as listed in Table 1, including 1) five knowledge-based IM methods: ICM (Pathak et al., 2017), RND (Burda et al., 2018), Disagreement (Pathak et al., 2019), MADE (Zhang et al., 2021), and AGAC (Flet-Berliac et al., 2021); 2) one data-based IM method called APT (Liu & Abbeel, 2021b); 3) and nine competence-based methods: DIAYN (Eysenbach et al., 2018), VISR (Hansen et al., 2019), DADS (Sharma et al., 2019), APS (Liu & Abbeel, 2021a), CIC (Laskin et al., 2022), MOSS (Zhao et al., 2022), BeCL (Yang et al., 2023), LSD (Park et al., 2021), and CSD (Park et al., 2023). For implementation details, please refer to Appendix B.

### 4.1.2 EXPERIMENTAL SETUP FOR EIM

**Environments.** We evaluate CIM for EIM in two navigation tasks in D4RL (Fu et al., 2020): PointMaze_UMaze-v3 and AntMaze_UMaze-v3. Note that CIM for EIM is orthogonal with any intrinsic bonuses. Unless otherwise mentioned, we adopt APT (Liu & Abbeel, 2021b), the state-of-the-art data-based IM method, to compute intrinsic bonuses in EIM tasks. The total instant reward is then $r = r_e + \tau_k^{\text{CIM}} r_i^{\text{APT}}$, where $r_e$ is the extrinsic reward and $r_i^{\text{APT}} := \log(1 + 1/k \sum_{j=1}^{k} \|\phi(s) - \phi(s)^j\|)$ is the intrinsic bonus.

**Baseline Methods and Implementation Details.** We compare CIM for EIM with other commonly used temperature schedule methods, including the constant schedule $\tau_k^{\text{C}} \equiv \beta_{\text{C}}$, the linear decay schedule $\tau_k^{\text{L}} = \beta_{\text{L}}(1 - k/T)$, and the exponential decay schedule $\tau_k^{\text{E}} = \beta_E(1 - \rho_{\text{E}})^k$. Please refer to Appendix B for more implementation details.

## 4.2 RESULTS IN RFPT TASKS

**Visualization of Skills** As previous works like LSD (Park et al., 2021) do, we train CIM for RFPT to learn diverse locomotion continuous skills in the Ant and Humanoid environment and diverse ma-

Table 2: State Coverage of 2D continuous locomotion or manipulation skills discovered by various typical IM methods.

| Environment | RND | APT | APS | CIC | LSD | CSD | CIM (ours) |
|---|---|---|---|---|---|---|---|
| Ant (29D) | $123 \pm 15$ | $33 \pm 3$ | $192 \pm 75$ | $697 \pm 200$ | $50 \pm 24$ | $4 \pm 0$ | $\mathbf{1042 \pm 158}$ |
| Humanoid (378D) | $22 \pm 1$ | $22 \pm 1$ | $107 \pm 33$ | $64 \pm 11$ | $8 \pm 1$ | $4 \pm 0$ | $\mathbf{1135 \pm 360}$ |
| FetchPush (25D) | $137 \pm 22$ | $\mathbf{154 \pm 17}$ | $79 \pm 14$ | $150 \pm 34$ | $24 \pm 12$ | $105 \pm 48$ | $141 \pm 15$ |
| FetchSlide (25D) | $182 \pm 52$ | $185 \pm 49$ | $178 \pm 33$ | $\mathbf{223 \pm 3}$ | $31 \pm 33$ | $114 \pm 79$ | $187 \pm 16$ |

Table 3: Ablation on the choice of the alignment loss $l_i$.

| Environment | $l_i^{\mathrm{MSE}}$ | $l_i^{\mathrm{vMF}}$ | $l_i^{\mathrm{LSD}}$ | $l_i^{\mathrm{CIC}}$ | $l_i^{\mathrm{BeCL}}$ | $l_i^{\mathrm{CIM}}$ (ours) |
|---|---|---|---|---|---|---|
| Ant (29D) | $64 \pm 20$ | $371 \pm 85$ | $28 \pm 14$ | $746 \pm 108$ | $726 \pm 70$ | $\mathbf{1042} \pm 158$ |

nipulation skills in FetchPush and FetchSlide. The learned skills are visualized as trajectories of the agent on the $x - y$ plane in Figure 1 and Figure 2. Our CIM for RFPT outperforms all 15 baselines in terms of skill diversity and state coverage. The skills learned via CIM are interpretable because of our alignment loss; the direction of the trajectory on the $x - y$ plane changes consistently with the change in the direction of the skill. Specifically, CIM excels at learning dynamic skills that move far from the initial location in almost all possible directions, while most baseline methods fail to discover such diverse and dynamic primitives. Their trajectories are non-directional or less dynamic than CIM, especially in two locomotion tasks. Competence-based approaches like DIAYN (Eysenbach et al., 2018), VISR (Hansen et al., 2019), and DADS (Sharma et al., 2019) directly maximize MI objectives but learn to take static postures instead of dynamic skills; such a phenomenon is also reported in LSD (Park et al., 2021) and CIC (Laskin et al., 2022). Although APS (Liu & Abbeel, 2021a) and CIC can learn dynamic skills by directly maximizing the state entropy, CIM discovers skills that reach farther and are more interpretable via maximizing the lower bound of the state entropy. As for the two variants of CIC, MOSS (Zhao et al., 2022) and BeCL (Yang et al., 2023), they perform even worse than CIC in all tasks, reflecting their limitation in skill discovery. Lastly, LSD (Park et al., 2021) and CSD (Park et al., 2023) cannot learn dynamic skills within limited environment steps in Ant and Humanoid due to their low sample efficiency. Though they perform better in manipulation tasks than locomotion tasks, their learned skills are rambling compared with our CIM.

**State Coverage**  To make a quantitative comparison between various IM methods, we measure their state coverage. The state coverage in Ant and Humanoid is determined by calculating the number of $2.5 \times 2.5 \ \mathrm{m}^2$ bins occupied on the x-y plane, based on 1000 randomly sampled trajectories. This was then averaged over five runs. For FetchPush and FetchSlide, we use smaller bins. As shown in Table 2, CIM significantly outperforms all the baseline methods in two torque-as-input locomotion tasks and is comparable in two position-as-input manipulation tasks. Although the state coverage of CIM is slightly lower than APT (Liu & Abbeel, 2021b) and CIC (Laskin et al., 2022) in FetchPush and FetchSlide, the skills learned via CIM are more interpretable, as shown in Figure 2.

**Fine-Tuning Efficiency in URLB**  We also evaluate CIM for RFPT in URLB (Laskin et al., 2021), a benchmark environment for RFPT in terms of fine-tuning efficiency. The results are presented in Table 4. The score (the last line of the table) is standardized by the performance of the expert DDPG, the same as in URLB and CIC (Laskin et al., 2022). CIM performs better in Run and Walk tasks and achieves the highest average score. The dynamic skills learned through CIM for RFPT can be adapted quickly to diverse fine-tuning tasks, including flipping and standing. Our experiments also show that the skill dimension $n_z = 3$ is better for CIM to discover flipping skills than $n_z = 2$. The fixed skill selection mechanism for CIM is the same as CIC.

**Ablation Study**  According to the results in Table 3, loss functions that follow the NCE style, such as $l_i^{\mathrm{CIC}}$, $l_i^{\mathrm{BeCL}}$, and $l_i^{\mathrm{CIM}}$, perform better than other styles like MSE and vMF. Besides, $l_i^{\mathrm{CIM}}$ is the most effective. As shown in Figure 3a, our CIM can also be utilized to discover discrete dynamic skills, though it is mainly designed for continuous skills. What more, our CIM for RFPT is also robust to the number of skill dimensions. The detailed results are provided in Appendix F.

Table 4: Fine-tuning average episode rewards $\pm$ standard deviations of eight methods in Walker domain of URLB. We denote knowledge-, data-, and competence-based methods in red, green, and blue, respectively. We bold the best result in each row. We report the normalized average score in the last row.

| Task | DDPG | RND | Proto | APS | CIC | MOSS | BeCL | CIM (ours) |
|------|------|-----|-------|-----|-----|------|------|-----------|
| Flip | $536 \pm 66$ | $470 \pm 47$ | $523 \pm 89$ | $407 \pm 104$ | $\mathbf{709 \pm 172}$ | $425 \pm 77$ | $628 \pm 46$ | $664 \pm 80$ |
| Run | $274 \pm 22$ | $403 \pm 105$ | $347 \pm 102$ | $128 \pm 38$ | $492 \pm 81$ | $244 \pm 13$ | $467 \pm 81$ | $\mathbf{585 \pm 27}$ |
| Stand | $931 \pm 18$ | $907 \pm 16$ | $861 \pm 79$ | $698 \pm 215$ | $939 \pm 28$ | $862 \pm 100$ | $\mathbf{951 \pm 3}$ | $941 \pm 21$ |
| Walk | $777 \pm 89$ | $844 \pm 99$ | $828 \pm 70$ | $577 \pm 133$ | $905 \pm 22$ | $684 \pm 40$ | $781 \pm 221$ | $\mathbf{921 \pm 30}$ |
| Score | $0.69 \pm 0.23$ | $0.72 \pm 0.20$ | $0.70 \pm 0.20$ | $0.49 \pm 0.25$ | $0.85 \pm 0.18$ | $0.60 \pm 0.22$ | $0.78 \pm 0.19$ | $\mathbf{0.86 \pm 0.11}$ |

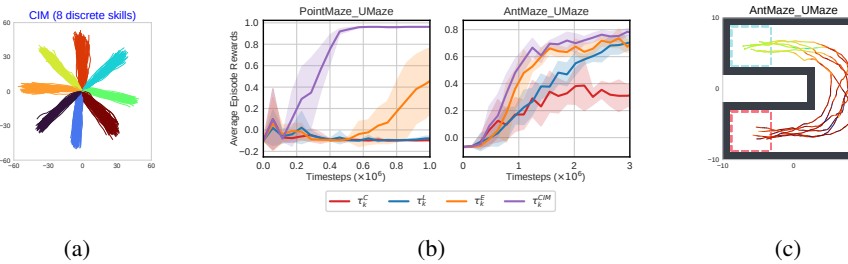

(a)           (b)           (c)

Figure 3: (a) Discrete CIM with $n_z = 8$ in Ant. The visualization in Humanoid is similar. (b) The learning curves of different temperature schedule methods. (c) Trajectory visualization of the meta-controller. The color of each sub-trajectory reflects the direction of the option.

### 4.3 RESULTS IN EIM TASKS

In PointMaze, we directly train a policy to control the Point without learning low skills since the dynamics is simple. In AntMaze, we train a meta-controller on top of the latent-conditioned policy pre-trained via our CIM for RFPT method. The meta-controller observes the target goal concatenated to the state observation $[s; s_g]$ and outputs the skill latent variable $z$ at each timestep. Figure 3b shows that the Lagrangian-based adaptive schedule $\tau_k^{\text{CIM}}$ outperforms other baseline methods, especially in PointMaze. Specifically, we can observe a small peak in the early stage of the training in PointMaze, which means the agent can reach the randomly generated target point with a small probability. However, as the training processes, the agent is distracted by the intrinsic bonuses when using the constant schedule $\tau_k^{\text{C}}$ or the linearly decayed schedule $\tau_k^{\text{L}}$ of the temperature. Note that $\tau_k^{\text{CIM}}$ is close to $\tau_k^{\text{E}}$ in AntUMaze, leading to similar performance curves. Moreover, other latent-conditioned policies are of poor quality, and we fail to train a mete-controller on top of these policies. We visualize the trajectories of the Ant in the $x - y$ plane as shown in Figure 3c, where the skills in a single trajectory gradually change to make the Ant turn a corner. We also conduct experiments to demonstrate the performance of CIM for EIM across four sparse-reward locomotion tasks. Please refer to Appendix F for details.

## 5 CONCLUSION

In this paper, we proposed Constrained Intrinsic Motivation (CIM) for RL. We designed a novel competence-based method for RFPT tasks to discover diverse and dynamic skills. This approach consisted of two key components: maximizing a lower bound of the conditional state entropy $H(s|z)$ and maximizing a lower bound of MI $I(s; z)$. Additionally, We designed an adaptive temperature schedule $\tau_k^{\text{CIM}}$ for EIM tasks based on constrained policy optimization. Our experiments demonstrated that our CIM for RFPT outperformed all baselines in multiple MuJoCo environments regarding diversity, state coverage, sample efficiency, and fine-tuning efficiency. Furthermore, the latent-conditioned policy learned via CIM for RFPT was successfully applied to complex EIM tasks via training a meta-controller on top of it. We also empirically verified the effectiveness of our schedule $\tau_k^{\text{CIM}}$ in multiple EIM tasks.

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

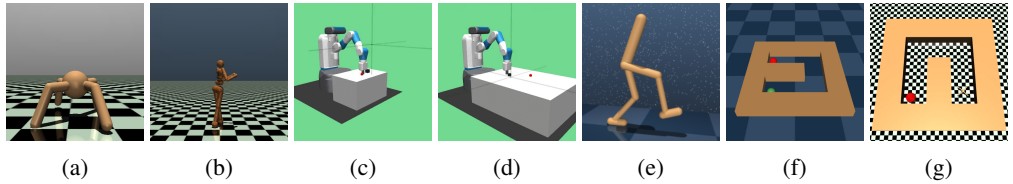

|     |     |     |     |     |     |     |
|-----|-----|-----|-----|-----|-----|-----|
| (a) | (b) | (c) | (d) | (e) | (f) | (g) |

Figure 4: (a)-(d) Ant, Humanoid, FetchPush and FetchSlide for evaluation of dynamic skill discovery; (e) Walker (Stand, Walk, Flip, Run) in URLB for evaluation of fine-tuning efficiency; and (f)&(g) PointMaze_UMaze and AntMaze_UMaze in D4RL for evaluation of CIM for EIM.

---

**Algorithm 1:** CIM for RFPT

---

Initialize the latent-conditioned policy $\pi$ and the state representation function $\phi$
**while** *not exceed* `total_timesteps` **do**
    **while** *not exceed* `steps_per_collect` **do**
        Sample skill $z$ from $p(z)$
        Collect samples $\{(s, a, s')\}$ with $\pi(a|s, z)$
    **end**
    Compute reward $r^{\text{CIM}}$ with Equation (8)
    Update $\phi$ using SGD to minimize the CIM alignment loss in Equation (6)
    Update $\pi$ using an on-policy method like PPO or an off-policy method like DDPG
**end**

---

## A  ENVIRONMENTS

To test the effectiveness of our CIM in high-dimensional tasks, we adopt Humanoid in Gymnasium MuJoCo environments. This environment has a 378D state space and a 17D action space, making it one of the most challenging environments available. We choose manipulation tasks to verify that our CIM can be utilized in environments other than locomotion tasks. Moreover, we evaluate the fine-tuning efficiency of the pre-trained skills in the URLB Walker domain, which has a 24D state space and a 6D action space. This domain includes four downstream tasks, i.e., Walker-Flip, Walker-Stand, Walker-Walk, and Walker-Run. Figure 4 shows all environments used in our experiments. The Ant and Humanoid environments are part of the MuJoCo environments in Gymnasium. FetchPush and FetchSlide are two tasks in the Fetch environment of the Gymnasium-Robotics, a collection of RL robotic environments. These four environments are used in previous skill discovery papers such as DIAYN, DADS, LSD, and CSD. In the URLB Walker domain, there are four downstream tasks: WalkerStand, WalkerFlip, WalkerWalk and WalkerRun. These tasks are designed to evaluate the fine-tuning efficiency of pre-trained latent-conditioned policies. To achieve a high average score, the unsupervised agent must discover diverse locomotion skills. In the PointMaze_UMaze environment, the Point (green) must navigate through the U-Maze from the initial region to a randomly generated target position (red). The target position is generated within a limited region, leading to a sparse reward. AntMaze_UMaze is more challenging than PointMaze_UMaze. The map of AntMaze_UMaze is 36× larger than PointMaze_UMaze.

## B  IMPLEMENTATION DETAILS

To maximize $J_i^{\text{CIM}}$ defined in Equation (4) and satisfy the alignment constraint at the same time, we alternatively train the policy $\pi$ via the on-policy RL algorithm Proximal Policy Optimization (PPO) and learn the state representation encoder $\phi$ via SGD. We implement all baselines in four Gymnasium environments using the same on-policy RL algorithm to ensure a fair comparison. All methods are trained with the same sampling budget of 40M environment steps (one-tenth of the original LSD) in Ant, 400M in Humanoid, and 4M in two Fetch tasks. In URLB, we implement our CIM based on their codes and follow the benchmark's standard training procedure to ensure a fair comparison, that is, pretraining the agent for 2M steps with only intrinsic rewards and then fine-tuning the pre-trained agent for 0.1M with only extrinsic rewards. Note that URLB adopts the

Table 5: Default key parameters of CIM for RFPT

| Parameter Name | Default value |
| --- | --- |
| total_timesteps | 4e7 |
| steps_per_collect | 512×64 |
| number_of_minibatches | 4 |
| number_of_training_envs | 64 |
| num_of_particles | 10 |

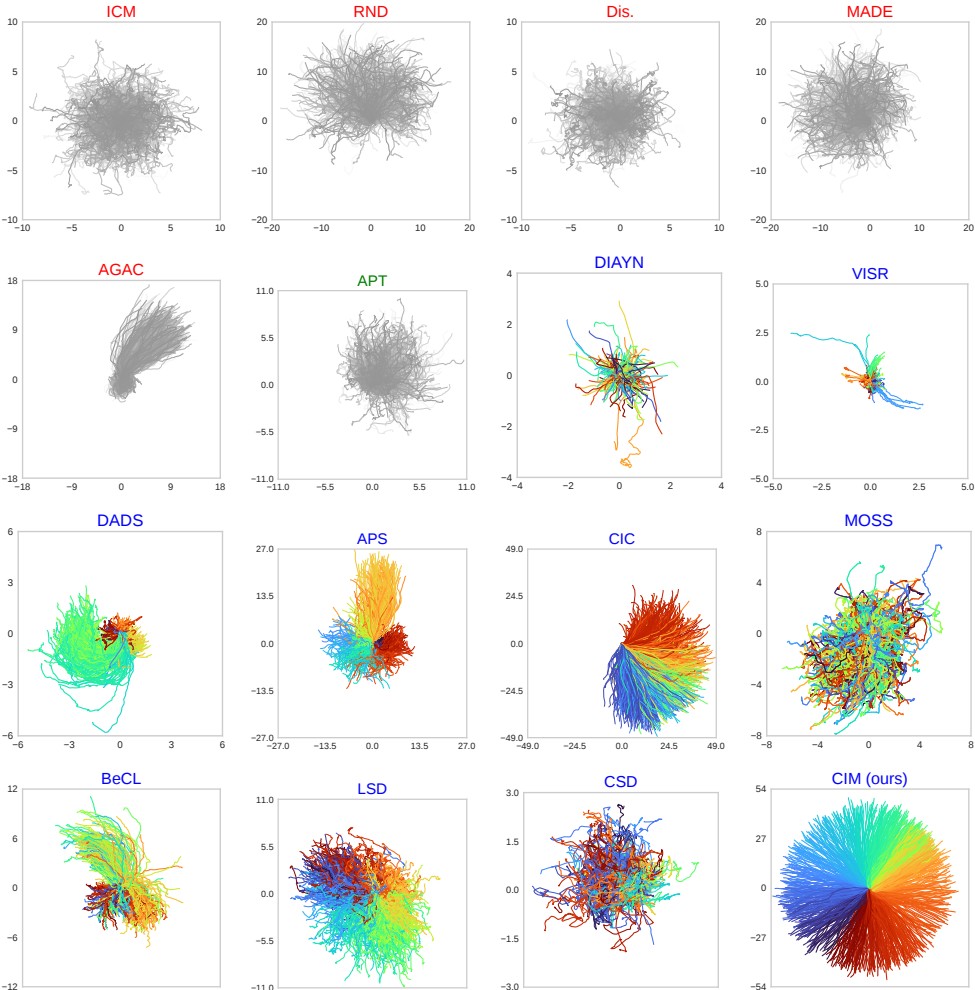

Figure 5: Enlarged Visualization of 2D continuous locomotion skills discovered by various IM methods in Ant. Each color in competence-based methods represents the direction of the skill latent variable $z$.

off-policy RL algorithm DDPG as the policy optimization method. All experiments are carried out across five random seeds in four Gymnasium environments and eight in URLB.

Algorithm 1 shows the training procedure of CIM for RFPT tasks. The state representation function $\phi$ and the latent-conditioned policy $\pi$ are updated alternately after steps_per_collect samples are sampled with randomly generated skill vectors. We fix the skill vector within a single episode as previous works do.

Table 6: Ablation on the number of dimensions of latent skill variables.

| Environment | 2 | 3 | 10 | 64 |
|---|---|---|---|---|
| Ant (29D) | **1042** $\pm$ 158 | $875 \pm 240$ | $901 \pm 20$ | $615 \pm 54$ |

Table 5 includes default key parameters of CIM for RFPT. CIM for RFPT and all baselines use the same parameters for the policy optimization method, i.e., PPO (in Gymnasium) or DDPG (in URLB), as in previous works.

## C    ENLARGED VISUALIZATION

Figure 5 shows the enlarged pictures of 2D continuous locomotion skills discovered by various IM methods in Ant.

## D    PROOF OF THEOREM 3

We provide the detailed proof for Theorem 3. Before moving on, we give the proof schetches for Proposition 1 and ƒ2 as follows.

Observing that $H(\mathrm{s}, g(\mathrm{s}))|\mathrm{z}=z) = H(\mathrm{s}|\mathrm{z}=z) + H(g(\mathrm{s})|\mathrm{s}, \mathrm{z}=z) = H(g(\mathrm{s})|\mathrm{z}=z) + H(\mathrm{s}|g(\mathrm{s}), \mathrm{z}=z)$, and $H(g(\mathrm{s})|\mathrm{s}, \mathrm{z}=z) = 0$ ($g$ is deterministic and $z$ is fixed), we have $H(\mathrm{s}|\mathrm{z}=z) - H(g(\mathrm{s})|\mathrm{z}=z) = H(\mathrm{s}|g(\mathrm{s}), \mathrm{z}=z) \geq 0$. If and only if $g$ is invertible ($H(\mathrm{s}|g(\mathrm{s}), \mathrm{z}=z) = 0$), we have $H(\mathrm{s}|\mathrm{z}=z) = H(g(\mathrm{s})|\mathrm{z}=z)$. Thus, Proposition 1 is proven. Further, based on Proposition 1 and the definition of the conditional entropy $H(\mathrm{s}|\mathrm{z}) = \mathbb{E}_{\mathrm{z}}[H(\mathrm{s}|\mathrm{z}=z)]$, Proposition 2 is proven.

We now present the proof of Theorem 3.

*Proof.* It is obvious that $g(s) = \max(\phi(s)^T, z)$ is a deterministic function. We then show that $g$ is invertible when the state representation $\phi(s)$ and the skill $z$ are well aligned (i.e., $\phi(s)^T z = \|\phi(s)\|\|z\|$). With $\phi(s)^T z = \|\phi(s)\|\|z\|$, we have $g = \max(\phi(s)^T z, 0) = \max(\|\phi(s)\|\|z\|, 0) = \|\phi(s)\|\|z\|$, which is an invertible function of $\phi(s)$ given a sampled $z$. This is because we can recover $\phi(s)$ from $g$, that is, $\phi(s) = \frac{g}{\|z\|} \frac{z}{\|z\|}$ with $\frac{g}{\|z\|}$ being the norm of $\phi(s)$ and $\frac{z}{\|z\|}$ being the direction of $\phi(s)$. Thus, according to Proposition 1 and 2, Theorem 3 is proven.    $\square$

## E    VENN DIAGRAM FOR RELATIONSHIPS INVOLVED IN IMRL

Since there are quite some Macros involved in IMRL, We illustrate their relationshipes in Figure 6. Specifically, IMRL can be divided into two branches, RFPT tasks and EIM tasks. URLB is a famous benchmark to evaluate the fine-tuning performance of IM methods for RFPT tasks. Note that there are three types of IM methods in previous works, i.e., knowledge-based, data-based, and competence-based. Our CIM for RFPT $J_i^{\mathrm{CIM}}$ belongs to competence-based IM methods. And our CIM for EIM is a novel adaptive temparerture scheduler $\tau_i^{\mathrm{CIM}}$.

## F    MORE EXPERIMENT RESULTS

### F.1    ABLATION ON THE NUMBER OF DIMENSIONS OF LATENT SKILL VARIABLES

We conduct ablation study on the number of dimensions of latent skill variables. Table 6 indicates that CIM for RFPT is robust to the number of skill dimensions regarding the state coverage in Ant. Interestingly, even when the number of dimensions of latent skill variables is larger than the the number of dimensions of the Ant's state vector, our CIM for RFPT can still learn dynamic skills to achieve large state coverage. On the other hand, for learning 2D locomotion skills, setting the number of dimensions of latent skill variables to 2 is enough.

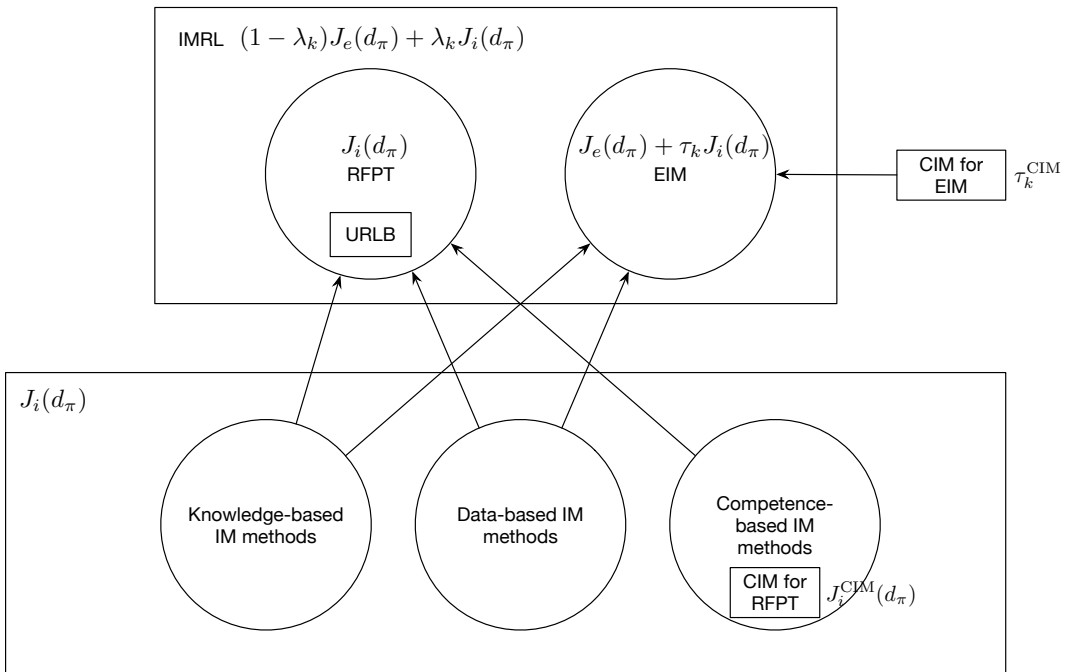

Figure 6: Venn diagram to explain the relation of IMRL, RFPT, EIM, URLB, CIM.

Table 7: The test-time average episode rewards $J_e(d_\pi)$ under different temperature schedulers.

| Temperature Scheduler | SHC | SA | SHS | SGW |
|---|---|---|---|---|
| $\tau_k^{\mathrm{C}}$ | 0.27 | 0.74 | 0.18 | 1 |
| $\tau_k^{\mathrm{CIM}}$ | **1** | **0.97** | **1** | **1** |

## F.2 MORE EXPERIMENT TO EVALUATE CIM FOR EIM

We chose two representative environments to illustrate the effectiveness of CIM for EIM in the main body of the paper due to space constraints. Here, we conduct more experiments to demonstrate the performance of CIM for EIM across four sparse-reward locomotion tasks (designed by Mutti et al., 2021): SparseHalfCheetah (SHC), SparseAnt (SA), SparseHumanoidStandup (SHS) and SpraseGridWorld (SGW). As shown in Table 7, $\tau_k^{\mathrm{CIM}}$ effectively reduces the bias introduced by intrinsic rewards, thereby enhancing performance in EIM tasks.

To show how our adaptive temperature scheduler $\tau_k^{\mathrm{CIM}}$ solves the issue of suboptimality, we plot the curves of $\tau_k^{\mathrm{CIM}}$ during training. From Figure 7, we can see that $\tau_k^{\mathrm{CIM}}$ gradually tends to zero across all EIM tasks. This means that our adaptive temperature $\tau_k^{\mathrm{CIM}}$ can effectively reduce the bias introduced by the intrinsic objective $J_i(d_\pi)$, i.e., the agent gradually focuses on maximizing only $J_e(d_\pi)$ to exploit the task rewards as $\tau_k^{\mathrm{CIM}}$ is near zero in the final stage of training, instead of being distracted by the intrinsic objective to do superfluous exploration.

## G COMPARISON WITH EXTRINSIC OPTIMALITY CONSTRAINT

Here, we compare our CIM for EIM with Chen et al. (2022)'s work. It should be noted that though our constraint in Equation (9) is similar to Extrinsic Optimality Constraint, $J_e(d_\pi) = \max_\pi J_e(d_\pi)$ proposed by Chen et al. (2022), there are some key differences. Firstly, we only need to train a single policy $\pi$, while they have to train two separate policies, $\pi_{e+i}$ and $\pi_e$, alternately. Secondly, we use an automated curriculum based on the agent's performance, as measured by $R_k$, to adjust the constraint strength. In contrast, their constraint is relatively harsh and difficult to satisfy during most of the training process.

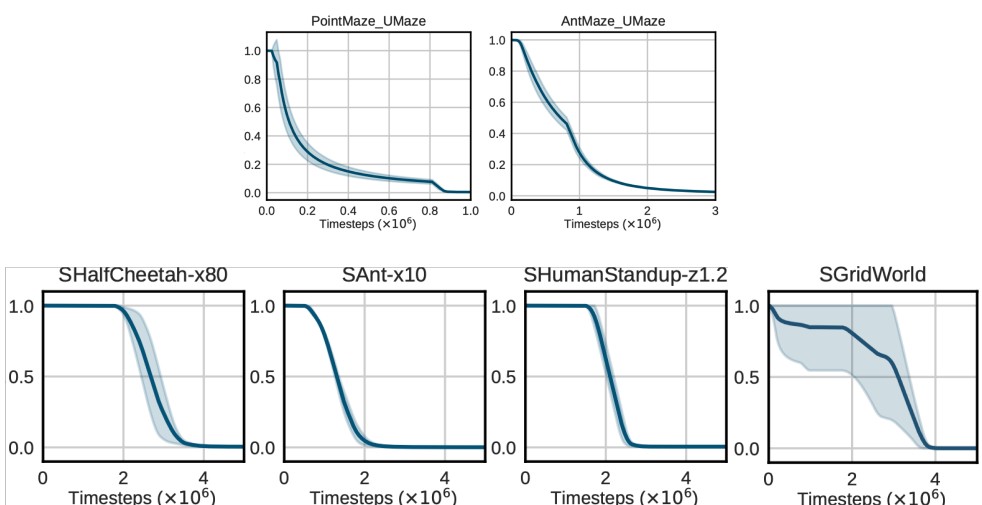

Figure 7: Curves of our adaptive temperature scheduler $\tau_k^{\text{CIM}}$ during training.

Table 8: List of Abbreviations of Common Items

| Abbreviations of Common Items | Full Name |
|---|---|
| EIM | Exploration with Intrinsic Motivation |
| IM | Intrinsic Motivation |
| IMRL | Intrinsically Motivated Reinforcement Learning |
| MDP | Markov Decision Process |
| MI | Mutual Information |
| RFPT | Reward-Free Pre-Training |
| RL | Reinforcement Learning |
| UCB | Upper Confidence Bound |
| URLB | Unsupervised Reinforcement Learning Benchmark |

# H LIST OF ABBREVIATIONS

We list all abbreviation of common items in Table 8 and all abbreviation of IM methods in Table 9.

Table 9: List of Abbreviations of IM Methods

| Abbreviations of IM Methods | Full Name |
| --- | --- |
| ICM (Pathak et al., 2017) | Intrinsic Curiosity Module |
| RND (Burda et al., 2018) | Random Network Distilling |
| Dis. (Pathak et al., 2019) | Disagreement |
| MADE (Zhang et al., 2021) | MAke DEviation from policy cover |
| AGAC (Flet-Berliac et al., 2021) | Adversarially Guided Actor-Critic |
| MaxEnt (Hazan et al., 2019) | Maximum state Entropy |
| APT (Liu & Abbeel, 2021b) | Active Pre-Training |
| RE3 (Seo et al., 2021) | Random Encoders for Efficient Exploration |
| VIC (Gregor et al., 2016) | Variational Intrinsic Control |
| DIAYN (Eysenbach et al., 2018) | Diversity Is All You Need |
| VISR (Hansen et al., 2019) | Variational Intrinsic Successor Feature |
| DADS (Sharma et al., 2019) | Dynamics-Aware unsupervised Discovery of Skills |
| APS (Liu & Abbeel, 2021a) | Active Pretraining with Successor Features |
| CIC (Laskin et al., 2022) | Contrastive Intrinsic Control |
| MOSS (Zhao et al., 2022) | a Mixture Of SurpriseS |
| BeCL (Yang et al., 2023) | Behavior Contrastive Learning |
| LSD (Park et al., 2021) | Lipschitz-constrained unsupervised Skill Discovery |
| CSD (Park et al., 2023) | Controllability-aware unsupervised Skill Discovery |
| CIM | Constrained Intrinsic Motivation |

