# OpenReview forum: "CIM: Constrained Intrinsic Motivation for Reinforcement Learning"
_ICLR.cc/2024/Conference — Submitted to ICLR 2024_

### Official Review · Reviewer_YFdU · 2023-10-30

**Soundness:** 3 good
**Presentation:** 2 fair
**Contribution:** 2 fair
**Rating:** 6
**Confidence:** 3

**Summary:**

As per the paper, there are two branches of intrinsic motivation RL (IMRL).
RFPT: Here, one expects to learn the occupancy measure, hence exploration is necessary.
EIM: Here, one wants to maximize the extrinsic objective, however, to do so agent also needs to explore (which is driven by intrinsic motivation). Hence the agent needs to balance the Intrinsic and extrinsic objectives with a $\tau$.

The author introduces Constrained Intrinsic Motivation (CIM) for RFPT and EIM. For RFPT, CIM maximizes a lower bound of the state entropy with a constraint that aligns with skill. For EIM, they propose a way to choose $\tau$.

**Strengths:**

- The paper is easy to understand
- Proposes a reward function that maximizes a lower bound of state entropy while aligning to skills
- CIM achieves good empirical performance

**Weaknesses:**

1. In the introduction, maybe focus early on
- the latent skill, why is it important
- what is the bias and what are its implications, e.g. why is it bad?

2. The work primarily focuses on reward shaping which can be considered a heuristic development (e.g., eqn 7) upon the prior works.
 However, the authors also tried to give intuitions defending their choices. I am unable to judge contribution/novelty here.

3. The presentation of theoretical results is not so sharp. Need to say about s,z,g. Is it true for any s,z, g? Are there some assumptions on g and what does "with equality" mean?

**Questions:**

On page 5, how are the last 4 lines related to eqn 10? I got a bit lost there.

What are typical latent skills? and how is policy conditioned on it? Is already apriori known which skills we want to learn?

There are quite some Macros, Maybe do images for e.g., with a Venn diagram to explain the relation of IMRL, RRFT, URLB, CIM, etc.

What is N in MI lower bound (above sec 3.1.2)

Typos: Page 5 Walfe -> Wolfe, sec 4.3 dynamic is -> dynamics is, Lemma 1 and Theorem 2 extra bracket ).

Comment: The paper studies an interesting problem of maximizing mutual information. It is related to experiment design kind of objectives or more widely maximizing submodular functions under MDP constraints. These functions capture that visiting the same state will result in reduced rewards and hence naturally encourage exploration to gain higher rewards. The submodular functions can thus encode state coverage, entropy maximization, and Mutual information (D-design) kind of objectives. There are works on submodular reinforcement learning that might be interesting and related to the authors.
The work is also related to convex RL but I see two papers cited (Hazan & Mutti) in that direction so I believe the authors are aware of it.

---

> ### Author Response · Authors · 2023-11-14
> **Author Response to Reviewer YFdU (Part 1/2)**
>
> We appreciate the reviewer’s insightful and constructive comments.
>
> **Q1: In the introduction, maybe focus early on the latent skill, why is it important, what is the bias and what are its implications.**
>
> A1: Thanks for the valuable suggestion. Following the reviewer's suggestion, we have discussed, in the Introduction section, the latent skill in competence-based IM methods in RFPT tasks and the bias introduced by knowledge- and data-based intrinsic rewards in EIM tasks in the revision. Especially, a policy conditioned on latent skill variables can consistently and meaningfully alter the environment's states, such as walking, jumping, flipping, or pushing. This makes it well-suited for efficient fine-tuning to solve downstream tasks. As for the bias introduced by intrinsic objectives in EIM tasks, intrinsic objectives can lead to superfluous exploration even when the task rewards are already accessible. This distraction, induced by intrinsic objectives, can deteriorate the performance of the agent. Consequently, it can impede the wider application of knowledge- and data-based IM methods.
>
> **Q2: The work primarily focuses on reward shaping which can be considered a heuristic development (e.g., eqn 7) upon the prior works. However, the authors also tried to give intuitions defending their choices. I am unable to judge contribution/novelty here.**
>
> A2: Please allow us to clarify that although the total reward $r_e + \tau_k r_i$ in EIM tasks can be viewed as reward shaping, our primary focus in RFPT tasks is on pre-training an agent without any task reward $r_e$ available, which does not fall into the domain of reward shaping since there is no task reward to shape. Actually, our CIM for RFPT is more related to unsupervised skill discovery. What's more, we design a novel alignment loss function (Eqn. (7)) based on CPC theory to maximize the MI between the state representation and the latent skill variable. We provide a thorough comparison with previous choices (Eqn. (6)) for the alignment loss function to highlight the distinction between our approach and prior works. We also conducted an ablation study on the choice of the alignment loss function to demonstrate its impact on the performance of CIM. As shown in Table 3, CIM for RFPT with our novel alignment loss function $l_i^{\text{CIM}}$ exhibits superior performance in Ant. Apart from maximizing the lower bound of $I(\rm{s};\rm{z})$ by minimizing the alignment loss defined in Eqn. (7), we also propose maximizing the lower bound of the conditional entropy $H(\rm{s}|\rm{z})$ via Eqn. (10). By alternately maximizing the lower bounds of both $I(\rm{s};\rm{z})$ and $H(\rm{s}|\rm{z})$, CIM for RFPT significantly improves the state coverage and promotes the discovery of dynamic and diverse skills compared with all 15 baseline methods, as shown in Fig. 1, Fig. 2, and Table 1. Besides state coverage and skill diversity, our CIM can also achieve a 90x faster pre-training speed than LSD and the highest fine-tuning score in the Walker domain of URLB.
>
> **Q3: The presentation of theoretical results is not so sharp.**
>
> A3: Thanks for the valuable comment. Lemma 1 and Theorem 2 is true for any $\rm{s},\rm{z}$, and any deterministic function $g(\rm{s})$. Please allow us to provide the detailed proof of Lemma 1 as follows to show why $g$ is assumed to be deterministic. As suggested in the latest ICLR latex template, we use $\rm{s}$ and $\rm{z}$ for variables and use $z$ for a concrete sample.
>
> *Proof of Lemma 1.*  Observing that $H(\rm{s},g(\rm{s}))|\rm{z}=$$z) = H(\rm{s}|\rm{z}=$$z) + H(g(\rm{s})|\rm{s},\rm{z}=$$z) = H(g(\rm{s})|\rm{z}=$$z) +  H(\rm{s}|g(\rm{s}),\rm{z}=$$z)$, and $H(g(\rm{s})|\rm{s},\rm{z}=$$z)=0$ ($g$ is deterministic), we have $H(\rm{s}|\rm{z}=$$z) - H(g(\rm{s})|\rm{z}=$$z) = H(\rm{s}|g(\rm{s}),\rm{z}=$$z) \ge 0$. If and only if $g$ is invertible ($H(\rm{s}|g(\rm{s}),\rm{z}=$$z) = 0$), we have $H(\rm{s}|\rm{z}=$$z) = H(g(\rm{s})|\rm{z}=$$z)$.
>
> Thus, the condition "with equality" means that $H(\rm{s}|\rm{z}=$$z) = H(g(\rm{s})|\rm{z}=$$z)$ if and only if we can deterministically infer $s$ given $g(s)$, i.e., $g$ is invertible, or in other words, $g$ is injective on the support of the state distribution $d_\pi(s)$. We have added the proof of Lemma 1 in Appendix F.2.

---

> ### Author Response · Authors · 2023-11-14
> **Author Response to Reviewer YFdU (Part 2/2)**
>
> **Q4: On page 5, how are the last 4 lines related to eqn 10?**
>
> A4: The last 4 lines are aimed at explaining how the intrinsic reward function can be derived from the convex intrinsic objective, i.e. the proof sketch for $r_i \propto \nabla_{d_\pi} L(d_\pi)$. The detailed proof is provided by Hazan via the Frank-Walfe algorithm. Based on this, we can directly derive the intrinsic reward of CIM $r_i^{\text{CIM}}$ for RFPT after designing the convex intrinsic objective of CIM $H(g(\rm{s})|z)$, that is, $r_i^{\text{CIM}} \propto \nabla_{d_\pi} H(g(\rm{s})|z) = \nabla_{d_\pi} (\sum_z\sum_s -d_\pi \log d_\pi) = -\log d_\pi - 1$. By applying the non-parametric $\rm{k}$-NN estimator, we obtain $r_i^{\text{CIM}} \propto - \log{1/\|g(s) - g(s)^\rm{k}\|}$, where $g(s)^\rm{k}$ stands for the $\rm{k}$-th nearest neighbour of $g(s)$. To ensure stability during the training process, we utilize the average version of $r_i^{\text{CIM}}$, as defined in Eqn. (10). This average version has been shown to be more stable for intrinsic reward estimation by previous works like APT (Liu & Abbeel 2021) and MOSS (Zhao et al. 2022).
>
> **Q5: What are typical latent skills? and how is policy conditioned on it? Is already apriori known which skills we want to learn?**
>
> A5: As answered in Q1, latent skills can be understood as the underlying abilities or action sequences that an agent can perform to consistently and meaningfully interact with the environment. For instance, as shown in Fig 1, the latent-conditioned policy learned via our CIM can control the Ant to run in any direction in the plane, which can be easily used for zero-shot goal-reaching tasks or as low-level skills for complex navigation tasks like AntUMaze. As stated in Section 3.1.2, following Eqn. (8), typically, the latent skill can be sampled from a prior distribution $p_z$. When there is no prior for the skills, a uniform distribution $\mathcal{U}^{n_z}(-1,1)$ is a common choice for sampling continuous skills, where $n_z$ is the skill dimension.
>
> To make the policy condition on the latent skill, we concatenate the current state and the sampled skill to form an input vector $[s,z]$ for the policy network $\pi(a|s,z)$.
>
> In this work, we do not rely on any prior knowledge or feature engineering for unsupervised skill discovery to make our algorithm more general.
>
> **Q6: There are quite some Macros, Maybe do images for e.g., with a Venn diagram to explain the relation of IMRL, RRFT, URLB, CIM, etc.**
>
> A6: Thanks for your valuable suggestion. We have drawn a Venn diagram to explain the relation of IMRL, RFPT, URLB, EIM, and CIM in Appendix E in the revision. Specifically, IMRL can be divided into two branches: RFPT tasks and EIM tasks. URLB is a famous benchmark to evaluate the fine-tuning performance of IM methods for RFPT tasks. Note that there are three types of IM methods in previous works, i.e., knowledge-based, data-based, and competence-based. Our CIM for RFPT $J_i^\text{CIM}$ belongs to competence-based IM methods. And our CIM for EIM is a novel adaptive temperature scheduler $\tau_i^\text{CIM}$.
>
> **Q7: What is N in MI lower bound (above sec 3.1.2)**
>
> A7: The $N$ stems from the Contrastive Predictive Coding (CPC) theory. Specifically, $N$ is the total number of samples for estimating the MI. Especially, this approximation becomes more accurate as $N$ increases; that is, the more negative samples, the more accurate the MI lower bound. We have added the definition of $N$ in the revision. Empirically, we found that $N=128$ is sufficient for our experiments.
>
> **Q8: Typos.**
>
> A8: Thanks for the valuable suggestions. We will fix the typos in the revision.
>
> **Q9: Relationships to submodular RL and convex RL.**
>
> A9: Thanks for your comments that our work is related to more broad optimization problems, including submodular RL and convex RL. This is the first time we heard of submodular RL, and we are happy to explore this promising domain in our future work. As you mentioned, we are aware of convex RL from Hazan and Mutti, and our work was originally inspired by Hazan, a data-based IM method that maximizes state entropy in the framework of convex RL.

---

> > ### Comment · Reviewer_YFdU · 2023-11-20
> >
> > Thanks to the Authors for responding. I have no more questions.

---

### Official Review · Reviewer_kU8G · 2023-10-30

**Soundness:** 3 good
**Presentation:** 3 good
**Contribution:** 3 good
**Rating:** 8
**Confidence:** 3

**Summary:**

The paper proposes a new constrained intrinsic motivation (CIM) objective to both maximize state coverage and distill skills, which solves limitations of coverage-based IM (like RND, not learning skills) and MI-based IM (like DIAYN, discouraging state coverage). It is examined on reward-free pre-training tasks and exploration using intrinsic motivation tasks. CIM works better than most baselines regarding state coverage and skills learning on various mujoco tasks.

**Strengths:**

1. Very exhaustive related work, good summary and comparison among them.
2. Baselines are exhaustive. Authors compared the proposed method with more than 10 baselines and outperform most of them, which is quite impressive.

**Weaknesses:**

Experiments performed for EIM tasks are a bit insufficient. Only on two environments, one directly trains using the proposed intrinsic reward while one trains a meta-controller on the top of learned skills. It would be more convincing to test on more environments.

**Questions:**

1. In section 4.2, State Coverage paragraph, APT and CIC outperform CIM on two tasks, FetchPush and FetchSlide. Do you have any intuition as to why these two methods are particularly better in these two tasks?
2. In Fig.3(a), in which task these discrete skills are learned? Also Ant or some other tasks?

Minior comments:
1. 5th line of page 2, “..., only maximizing MI only…”, one of “only”s should be removed?
2. S_T, s’, z from table 1 are not explained in the caption, although they are here and there in the paper, I do think it would be clearer to add them in the caption as well.

---

> ### Author Response · Authors · 2023-11-14
> **Author Response to Reviewer kU8G**
>
> We thank the reviewer for the valuable comments.
>
> **Q1: Additional Experiments for EIM tasks.**
>
> A1: We chose two representative environments to illustrate the effectiveness of CIM for EIM in the main body of the paper due to space constraints. Here, we conduct more experiments to demonstrate the performance of CIM for EIM across four sparse-reward locomotion tasks (designed by Mutti et al., 2021): SparseHalfCheetah (SHC), SparseAnt (SA), SparseHumanoidStandup (SHS) and SpraseGridWorld (SGW). The test-time average episode rewards $J_e(d_\pi)$ are reported in the table below:
>
> | **Scheduler** | **SHC** | **SA** | **SHS** | **SGW** |
> |---|---|---|---|---|
> |$\tau_k^{\text{C}}$ | 0.27 | 0.74 | 0.18 | 0.01 |
> |$\tau_k^{\text{CIM}}$ | **1** | **0.97** | **1** | **1** |
>
> As shown in the table, $\tau_k^{\text{CIM}}$ effectively reduces the bias introduced by intrinsic rewards, thereby enhancing performance in EIM tasks. We also draw the curves of $\tau_k^{\text{CIM}}$ in all EIM tasks to demonstrate that $\tau_k^{\text{CIM}}$ gradually tends to zero in Appendix F.2.
>
> **Q2: Intuition as to why APT and CIC outperform CIM regarding state coverage on FetchPush and FetchSlide.**
>
> A2: FetchPush and FetchSlide are position-as-input manipulation tasks that feature less complex environmental dynamics compared to the torque-as-input locomotion tasks Ant and Humanoid. As a result, both the data-based IM method APT (Liu & Abbeel 2021) and the competence-based methods CIC (Laskin et al. 2022), CSD (Park et al. 2023), and CIM can maximize the state coverage with only 4M samples (5x less than Ant). However, the skills learned via our CIM are more interpretable than those acquired through APT and CIC. As depicted in Fig. 2, the trajectory's direction on the $x−y$ plane consistently changes in line with the skill direction (observed as a smoother color transition). This interpretation can also be seen in Fig 1. We have added "position-as-input" and "torque-as-input" in the revision to imply the complexity of dynamics in these tasks.
>
> **Q3: In Fig.3(a), in which task these discrete skills are learned? Also Ant or some other tasks?.**
>
> A3: In Fig. 3(a), we present a visualization of the discrete skills learned using CIM for RFPT in the Ant task. The visualization of the discrete skills learned in Humanoid is similar.
>
> **Q4: Minor comments.**
>
> A4: Thanks for your valuable suggestions. We will revise our paper accordingly. We apologize for any confusion caused by the omission of the definitions for $s_T$, $s'$, and $z$ in Table 1 due to page limits. Specifically, $s_T$ represents the last state in one trajectory, where $T$ denotes the final time step in one episode. $s'\sim P(s'|s,a)$ is the subsequent state transitioned from the current state $s$ when action $a$ is taken. Lastly, $z$ is the latent skill. Following your suggestion, we have included these definitions in the caption of Table 1 to improve clarity.

---

> > ### Comment · Reviewer_kU8G · 2023-11-21
> > **Thanks for your response**
> >
> > Thank you very much for the clarification, I don't have further questions and will maintain the score I had.

---

### Official Review · Reviewer_hXz5 · 2023-10-31

**Soundness:** 2 fair
**Presentation:** 1 poor
**Contribution:** 3 good
**Rating:** 3
**Confidence:** 4

**Summary:**

The paper proposed a constrained intrinsic motivation (CIM) for the exploration of Reinforcement Learning. The claimed advantage of CIM is that it reduces the bias introduced for exploration with intrinsic motivation (EIM). CIM combined the advantage of data-based IM and the mutual information based IM. The simulations on various robotic control environments demonstrate the advantage of the proposed methods in terms of skill diversity and sample efficiency.

**Strengths:**

I believe the proposed form of intrinsic motivation is novel and a good way to combine the benefits of max-entropy exploration and mutual information-based exploration that generally leads to a diverse skill set.

The paper did comprehensive simulations studies with detailed comparisons between previous IM methods in the literature and illustrative visualizations that help the audience access the advantage of the proposed method.

**Weaknesses:**

The paper is hard to follow. Part of the reason is that there are too many abbreviations of previous approaches. I have to go back and forth to make sure I do not miss important information. I think the authors should replace them with in-text citation if possible. In general, I understand that good empirical results themselves can make a good paper. However, the use of notations and formulas should help to clarify the high-level idea and reduce the ambiguity. However, I found it hard to follow the notations and they are not self-contained in this paper and some formulas can be extremely confusing. I am listing a number of examples here.

1. We should not call Theorem 2 a theorem as it is a text-book level property in information theory, which should not considered as a contribution of the paper

2. (5), despite being the main contribution of the paper, is be very confusing. The constraint seems to be on the function $\phi$. However, the intrinsic objective does not depend on $\phi$. Should the constraint be on the learned skill $z$ such that there exists a representation of state that leads to good alignment. In such case, the constraint should be $\exists \phi, L_a(\phi(s), z) \leq c$, where $c$ is some constant. I think the definition of the CIM really has to be made clear as it is the main contribution of the paper.

3. Right above (10), the paper mentioned that the bound is tight when \phi(s) and z is well aligned. I don’t think this is correct. The function $g$ is already not invertible, so $H(g(s) \mid z)$ bound is never tight.

4. What is $k$ in the sentence “the intrinsic reward of CIM for RFPT is then ….”?

5. In (11), I don’t see how this constraint optimization problem solve the issue of suboptimality. This basically ensures that the value of the online-policy is non-decreasing if we consider the approximation of $\hat R_{k}$ you mentioned in the following paragraph. This does not guarantee that the policy can always be optimized to the true optimal policy.


I personally think that the proposed methods (if I understand it correctly) is a very novel approach and the simulation studies are interesting and complete. However, I don't think this paper is ready to publish unless the writing is significantly improved.

**Questions:**

1. Could the authors explain what it means by having bias for exploration, which occurs many times in Introduction? By bias, we typically mean that some estimator is inconsistent. It is not clear what it means by bias for exploration.
2. You mentioned that knowledge-based and data-based IM introduces non-negligible bias. Why doesn’t skill-based exploration introduce bias? In a EIM setting, it seems that all IM methods with non-decreasing temperature introduce bias.

---

> ### Author Response · Authors · 2023-11-14
> **Author Response to Reviewer hXz5 (Part 1/3)**
>
> We appreciate the reviewer's thoughtful and constructive feedback.
>
> ### Abbreviations
>
> > The paper is hard to follow. Part of the reason is that there are too many abbreviations of previous approaches. I have to go back and forth to make sure I do not miss important information. I think the authors should replace them with in-text citation if possible.
>
> Thanks for your helpful suggestion. We will revise our paper accordingly. To thoroughly summarize and compare previous Intrinsic Motivation (IM) methods, we had to include all 15 methods, which resulted in numerous abbreviations. As the reviewer suggested, we have replaced them with in-text citations in the revision to improve the readability of our paper.
>
> ### Notations and formulas
>
> > In general, I understand that good empirical results themselves can make a good paper. However, the use of notations and formulas should help to clarify the high-level idea and reduce the ambiguity. However, I found it hard to follow the notations and they are not self-contained in this paper and some formulas can be extremely confusing.
>
> Thanks for recognizing the novelty of our work and the strong empirical results. We also appreciate your valuable suggestions regarding our use of notations and formulas. We hope the following clarifications can help address your concerns about the notations and formulas.
>
> **Q1: We should not call Theorem 2 a theorem as it is a text-book level property in information theory, which should not considered as a contribution of the paper.**
>
> A1: We apologize for any confusion here. Although Theorem 2 is a textbook-level property, to the best of our knowledge, it hasn't been formally presented in any textbooks, which might not be easy to derive or understand for readers unfamiliar with information theory. Considering its importance for deriving the lower bound of the conditional state entropy, we decided to introduce it as a theorem. Following the reviewer's suggestion, we have changed Theorem 2 to Lemma 2 in the revised paper.
>
> Although the property itself is not our contribution, its application in the unsupervised skill discovery domain to derive the lower bound of the conditional state entropy $H(\rm{s}|\rm{z})$ is one of our main contributions. Previous IM methods like APT (Liu & Abbeel 2021) and CIC (Laskin et al. 2022), which directly maximize the state entropy $H(\rm{s})$, exhibit poor performance in state coverage and skill diversity, as shown in Fig. 1 and Table 2. By indirectly maximizing the state entropy $H(\rm{s})=H(\rm{s}|\rm{z})+I(\rm{s};\rm{z})$ through alternating maximization of the lower bounds of $H(\rm{s}|\rm{z})$ and $I(\rm{s};\rm{z})$, our CIM for RFPT significantly outperforms all 15 baseline methods in RFPT tasks. Moreover, our CIM for RFPT is 90x faster than LSD (Park et al. 2021) and CSD (Park et al. 2023) thanks to our novel lower bound of the conditional state entropy for unsupervised skill discovery.
>
> **Q2: Eqn. (5), despite being the main contribution of the paper, is be very confusing. The constraint seems to be on the function $\phi$. However, the intrinsic objective does not depend on $\phi$.**
>
> A2: There might be a misunderstanding. The constraint is indeed on the function $\phi$ and the intrinsic objective does depend on $\phi$. In fact, competence-based IM methods, e.g., APS (Liu & Abbeel 2021a), CIC (Laskin et al. 2022), MADE (Zhao et al. 2022), typically use the notation $H(\rm{s})$ to stand for $H(\phi(\rm{s}))$ since the state entropy is estimated in the state projection space $\mathcal{Z}$ instead of the original state space $\mathcal{S}$ due to the potential high dimensionality of the original state space, such as 378 in Humanoid. For pixel-based tasks, estimating the state entropy directly in the pixel space could be infeasible. We follow this tradition in the IM domain. Specifically, we also estimate the conditional state entropy in the state projection space and use $H(\rm{s}|\rm{z})$ as an abbreviation for $H(\phi(\rm{s})|\rm{z})$. Therefore, the intrinsic objective $H(\rm{s}|\rm{z})$ (i.e., $H(\phi(\rm{s})|\rm{z})$) actually depends on $\phi: \mathcal{S}\to \mathcal{Z}$. We have added detailed explanation for Eqn. (5) to clarify the dependence of $H(\rm{s}|\rm{z})$ on $\phi$ in the revision. Table 3 shows that the learned state representation $\phi(s)$ in the projection space can significantly impact the optimization of $H(\rm{s}|\rm{z})$. Additionally, in the IM domain, $z$ is typically sampled from a pre-defined distribution, such as the uniform distribution $\mathcal{U}^{n_z}(-1,1)$ for continuous skills when there is no prior knowledge ($n_z$ is the skill dimension). Hence, there is no constraint on $z$ typically. Furthermore, given a certain distribition for $z$, our constraint $\phi(s)\in\arg\min L_a(\phi(s),z)$ is equialent to $\exists \phi, L_a(\phi(s),z)\le c$ when setting $c$ to the minimum of $L_a(\phi(s),z)$, i.e., $c = \min_\phi L_a(\phi(s),z)$.

---

> ### Author Response · Authors · 2023-11-14
> **Author Response to Reviewer hXz5 (Part 2/3)**
>
> **Q3: Right above (10), the paper mentioned that the bound is tight when \phi(s) and z is well aligned. I don’t think this is correct. The function $g$ is already not invertible, so $H(g(s)|z)$ bound is never tight.**
>
> A3: Please allow us to clarify that the function $g=\max(\phi(s)^Tz, 0)$ is invertible when the state representation $\phi(s)$ and the skill $z$ are well aligned. The state representation $\phi(s)$ and the skill $z$ being well aligned means that the state representation and the skill are in the same direction, that is, $\phi(s)^Tz=\|\phi(s)\|\|z\|$. Therefore, when the state representation $\phi(s)$ and the skill $z$ are well aligned, $g=\max(\phi(s)^Tz,0)=\max(\|\phi(s)\|\|z\|, 0)=\|\phi(s)\|\|z\|$, which is an invertible function of $\phi(s)$ given a certain $z$. This is because we can recover $\phi(s)$ from $g$, that is, $\phi(s) = \frac{g}{\|z\|}\frac{z}{\|z\|}$ with $\frac{g}{\|z\|}$ being the norm of $\phi(s)$ and $\frac{z}{\|z\|}$ being the direction of $\phi(s)$. Thus, according to Lemma 1 and Theorem 2, the lower bound $H(g(s)|z)$ is tight when the state representation $\phi(s)$ and the skill $z$ are well aligned. We have formalized this argument as Theorem 3 in the revision and provided the proof in Appendix D.
>
> **Q4: What is $\rm{k}$ in the sentence ``the intrinsic reward of CIM for RFPT is then..."?**
>
> A4: In the sentence "the intrinsic reward of CIM for RFPT is then...", we use $\rm{k}$ to denote the $\rm{k}$−nearest neighbor, that is, in $r_i^\text{CIM}(s)=\|g(s)-g(s)^\rm{k}\|$, $g(s)^\rm{k}$ is the $\rm{k}$-th nearest neighbor of $g(s)$ according to Eqn. (9). To clarify the notation, we distinguished between the $\rm{k}$ and $k$ when we introduced the $\rm{k}$-NN estimator (Eqn. (9)). Specifically, $k$ represents the $k$-th iteration of policy optimization in RL, while $\rm{k}$ is the $\rm{k}$−nearest neighbor in $\rm{k}$-NN estimator. If it would be clearer, we could use $\kappa$ or $K$ for $\rm{k}$−NN estimator in the revision. We have added this description after the sentence "the intrinsic reward of CIM for RFPT is then..." to make it clear.
>
> **Q5: In (11), I don’t see how this constraint optimization problem solve the issue of suboptimality.**
>
> A5: As discussed in Section 3.2, we leverage the Lagrangian method to convert Eqn. (11) to $\min_{\lambda\le 0}\max_{d_\pi} J_e(d_\pi) + \lambda^{-1} J_i(d_\pi)$, where $\lambda$ is updated via SGD. As long as $\lambda^{-1}$ gradually tends to zero, the issue of suboptimality is solved.  Essentially, the agent gradually focuses on maximizing only $J_e(d_\pi)$ instead of $J_e(d_\pi) + J_i(d_\pi)$ in the final stage of training. Our empirical studies showed that $\tau_k^{\text{CIM}}=\min\\{\lambda_k^{-1}, 1\\}$ can gradually decrease from the initial value 1 to a near-zero number (e.g., $1e-4$ in AntUMaze) throughout the training process. To demonstrate how the issue of suboptimality is solved, we have conducted an additional experiment in extra EIM tasks (designed by Mutti et al. 2021), including SparseHalfCheetah (SHC), SparseAnt (SA), SparseHumanoidStandup (SHS) and SpraseGridWorld (SGW). The test-time average episode rewards $J_e(d_\pi)$ are as follows:
>
> | **Scheduler** | **SHC** | **SA** | **SHS** | **SGW** |
> |---|---|---|---|---|
> |$\tau_k^{\text{C}}$ | 0.27 | 0.74 | 0.18 | 0.01 |
> |$\tau_k^{\text{CIM}}$ | **1** | **0.97** | **1** | **1** |
>
> Our adaptive temperature $\tau_k^{\text{CIM}}$ can effectively reduce the bias introduced by the intrinsic objective $J_i(d_\pi)$, i.e., the agent gradually focuses on maximizing only $J_e(d_\pi)$ to exploit the task rewards as $\tau_k^{\text{CIM}}$ is near zero in the final stage of training, instead of being distracted by the intrinsic objective to do superfluous exploration. We have appended the curves of $\tau_k^{\text{CIM}}$ in all EIM tasks in Appendix F.2.

---

> ### Author Response · Authors · 2023-11-14
> **Author Response to Reviewer hXz5 (3/3)**
>
> ### Bias Introduced by Knowledge- and Data-based IM Methods in EIM Tasks
>
> **Q1: Could the authors explain what it means by having bias for exploration, which occurs many times in Introduction? By bias, we typically mean that some estimator is inconsistent. It is not clear what it means by bias for exploration.**
>
> A1: In response to your question, it's important to distinguish between two types of biases discussed in the Introduction. The first type is the inductive bias in RFPT tasks, such as the $x-y$ prior utilized by DIAYN (Eysenbach et al. 2018) and DADS (Sharma et al. 2019) for RFPT tasks. This form of inductive bias can be regarded as feature engineering (Park et al. 2021). As LSD (Park et al. 2021) and CSD (Park et al. 2023), we don't rely on feature engineering when designing CIM for RFPT to ensure the generality of our methods.
>
> The second type of bias is the bias introduced by the intrinsic objective in EIM tasks. As outlined in Section 2, in EIM tasks, the agent commonly employs a knowledge- or data-based IM method to stimulate exploration. This is because the task reward is typically sparse and hard to obtain at the start of training. The agent's goal is thus to maximize $J_e(d_\pi) + \tau_k J_i(d_\pi)$ (Eqn. (3)). However, the intrinsic objective $J_i(d_\pi)$ can distract the agent from maximizing the evaluation metric $J_e(d_\pi)$. Specifically, the optimal policy for maximizing $J_e(d_\pi) + \tau_k J_i(d_\pi)$ might not be the optimal policy for maximizing $J_e(d_\pi)$ when the agent is distracted by the intrinsic objective $J_i(d_\pi)$ to collect intrinsic rewards instead of obtaining the extrinsic rewards. Such a distraction phenomenon has also been reported in previous works (Chen et al. 2022). Chen et al. name this bias "intrinsic reward bias". Therefore, we did not use the term "bias for exploration" to describe this type of bias in EIM tasks. Instead, we refer to the "bias" in EIM tasks as the negative influence $J_i(d_\pi)$ exerts on the performance of the agent, the same as Cen et al..
>
> Sorry for the confusion caused by "how to reduce the bias introduced by the intrinsic objective for Exploration with Intrinsic Motivation (EIM)" in the Abstract. We have revised it as "how to reduce the bias introduced by the intrinsic objective in Exploration with Intrinsic Motivation (EIM) tasks".
>
> **Q2: You mentioned that knowledge-based and data-based IM introduces non-negligible bias. Why doesn’t skill-based exploration introduce bias? In a EIM setting, it seems that all IM methods with non-decreasing temperature introduce bias.**
>
> A2: We concur with your perspective that skill-based exploration could potentially introduce bias in EIM tasks. However, as stated in Section 2, to the best of our knowledge, all previous related studies primarily employ knowledge- or data-based exploration methods in EIM tasks, rather than skill-based exploration methods. We thus only mention that knowledge- and data-based IM introduces non-negligible bias in EIM tasks. We hypothesize that one key reason for this is the complexity of simultaneously discovering skills using a skill-based exploration method and maximizing the extrinsic objective $J_e(d_\pi)$ in the sparse-reward EIM tasks. We are willing to explore EIM with skill-based exploration in our future work.

---

> > ### Comment · Reviewer_hXz5 · 2023-11-20
> >
> > Thanks for the responses.
> >
> > Q1: I suggest using Proposition instead of Lemma because Lemmas are typically what we use to show main Theorem.
> >
> > Q2: I strongly suggest the authors to use the full notation $H(\phi(s) \mid z)$ in Equation (5), then introduce the abbreviation. In the current form, it is extremely confusing. With this in place, (5) is still confusing. Could you make it explicit about what variables we are optimizing when we optimize $H(\phi(s) \mid z)$? I suspect here that we are optimizing a policy, which induces the state distribution.
> >
> > Q3: the function $f(x) = \max\{x, 0\}$ is already not an invertible function. You cannot recover $x$ when $x$ is negative.
> >
> > Q4: could you use different notations for the two k's? It is hard for the reader to even notice this difference.
> >
> > Q5: could you make this explicit in (11)? The Lagrangian that is exactly equivalent to (11) has a non-decreasing $\lambda$ because $R_k$ is non-decreasing. By having a scheduled $\lambda$'s, you are essentially having a fixed schedule of $R_k$'s instead of setting $R_k$ to be the expected reward at the $k$-th. This is why it is confusing to the readers.
> >
> > I appreciate the authors' efforts in answering my questions, but I still think the paper needs significant polishing before it is ready to publish and I decided to keep my rating.

---

> > > ### Author Response · Authors · 2023-11-22
> > > **Author Response to Reviewer hXz5**
> > >
> > > Thank you very much for the timely and constructive feedback on our responses. We are so glad that the previous clarifications are helpful in addressing some of your concerns, and greatly appreciate your new comments/suggestions.
> > > We have further updated our work based on the new comments; please kindly let us know if anything is still not clear or should be revised.
> > >
> > > **Q1: I suggest using Proposition instead of Lemma because Lemmas are typically what we use to show main Theorem.**
> > >
> > > A1: Thanks for your valuable suggestion. We have changed the two Lemmas into two Propositions in our latest revision.
> > >
> > > **Q2: I strongly suggest the authors to use the full notation $H(\phi(\rm{s})|\rm{z})$ in Equation (5), then introduce the abbreviation. In the current form, it is extremely confusing. With this in place, (5) is still confusing. Could you make it explicit about what variables we are optimizing when we optimize $H(\phi(\rm{s})|\rm{z})$? I suspect here that we are optimizing a policy, which induces the state distribution.**
> > >
> > > A2: Thanks for the valuable suggestion. We agree with the reviewer that using the full notation $H(\phi(\rm{s})|\rm{z})$ is more friendly for readers unfamiliar with unsupervised skill discovery. Following your suggestion, we have replaced $H(\rm{s}|\rm{z})$ with $H^\pi(\phi(\rm{s}|\rm{z}))=E_zE_{\phi(s)\sim d_{\pi(\cdot|s,z)}} [-\log d_{\pi(\cdot|s,z)}]$ and made the optimization variable $\pi$ explicit in Eqn. (5). We have updated the changes to the latest version of our paper.
> > >
> > > **Q3: the function $f(x)=\max(x,0)$ is already not an invertible function. You cannot recover $x$ when $x$ is negative.**
> > >
> > > A3: Please allow us to clarify that: 1) Propositions 1 and 2 state that the **equality** is achieved iff $g$ is invertible; and 2) Theorem 3 states that the **equality** is achieved iff $\phi(s)^Tz=\|\phi(s)\|\|z\|\ge0$, in which case, $g$ becomes an invertible function $g=\phi(s)^Tz$. Once the alignment loss $L_a$ is minimized, this condition can be achieved. Please also note that the condition (well-aligned trajectory $\phi(s)$ and skill vector $z$) is also empirically verified in our experiments (see the straight trajectories in `CIM (ours)', Figure 1).
> > >
> > >
> > > **Q4: could you use different notations for the two k's? It is hard for the reader to even notice this difference.**
> > >
> > > A4: Thanks for the thoughtful comment. We have changed the $\rm{k}$ in the $\rm{k}$-nearest neighbor algorithm to $\xi$ , i.e.,  the $\xi$-th nearest neighbor.
> > >
> > > **Q5: could you make this explicit in (11)? The Lagrangian that is exactly equivalent to (11) has a non-decreasing $\lambda$ because $R_k$ is non-decreasing. By having a scheduled $\lambda$'s, you are essentially having a fixed schedule $R_k$ of 's instead of setting $R_k$ to be the expected reward at the $k$-th. This is why it is confusing to the readers.**
> > >
> > > A5: There might be a misunderstanding. Please note that in Eqn. (11) (now is Eqn. (10) in the updated version), it is $\lambda^{-1}$ (not $\lambda$), meaning that the $\tau_k^\text{CIM} = \min (\lambda_k^{-1}, 1)$ gradually tends to zero as $\lambda_k$ grows. In terms of the relationship between $\lambda_k$ and $R_k$, $\lambda_k$ is updated by SGD, i.e., $\lambda_k = \lambda_{k-1} - \eta ( J_e(d_{\pi_k}) - R_{k-1})$.  Thus, $\lambda_k$ is not a fixed scheduler since both $ J_e$ and $R_{k-1}$ depend on the agent's learning process. This is one of our contributions. This is also empirically verified in Fig. 7 (in Appendix G), where it shows adaptively decreased $\tau_k^\text{CIM}$ (the inverse of $\lambda_k$).
> > >
> > > Thank you again for the valuable suggestions. We are very fortunate to further improve the paper with your help.  It is not easy to put all the full/detailed names, notations, derivations, and descriptions into this paper, as we have investigated, summarized, implemented, and compared 15 intrinsic motivation methods for reinforcement learning. We understand that there is still much room for improvement and will work our best to achieve that. We sincerely thank the reviewer for the two rounds of discussions, and would greatly appreciate it if the reviewer could re-assess our work based on the addressed concerns and revised manuscript. This would mean a lot to us. Thank you very much!

---

### Official Review · Reviewer_9QWm · 2023-11-02

**Soundness:** 3 good
**Presentation:** 3 good
**Contribution:** 3 good
**Rating:** 6
**Confidence:** 3

**Summary:**

The paper focuses on two problems: reward-free exploration in RL and reducing the bias caused by intrinsic motivation. For the first problem, the proposed method CIM, solves a constrained maximization of a state entropy lower bound. The constraints encourage skill discovery and state coverage. Furthermore, CIM adaptively adjusts the intrinsic motivation strength to reduce the bias caused by the intrinsic reward. The proposed approach has a better sample efficiency on MuJuCo tasks.

**Strengths:**

- Empirically, CIM has a better sample efficiency and state coverage, especially in certain environments such as Ant.
- The proposed approach has significantly better skills discovery compared to prior methods.
- The authors propose a scheduling technique that effectively reduces intrinsic motivation bias.

**Weaknesses:**

- Novelty is limited; the algorithm mainly combines two algorithmic approaches in exploration: increasing state coverage and skill discovery.

**Questions:**

Comments:

- I’m not sure the knowledge-based and data-based classification of methods in the introduction is entirely accurate. For example, the objective given in Zhang et al. 2021 can recover maximum entropy exploration technique in special case.
- $\tau_k$ is more like a hyperparameter to control the trade-off between exploration and exploitation and is not really a temperature parameter—those are usually used in the form $\exp(\tau x)$ such as in softmax outputs or Boltzmann exploration.

---

> ### Author Response · Authors · 2023-11-14
> **Author Response to Reviewer 9QWm**
>
> We appreciate the reviewer's valuable comments.
>
> **Q1. Novelty is limited; the algorithm mainly combines two algorithmic approaches in exploration: increasing state coverage and skill discovery.**
>
> A1: While our algorithm may seem like a combination of data- and competence-based Intrinsic Motivation (IM) methods, the techniques and optimization procedures proposed in this work differ significantly from previous methods. Specifically, we don't directly maximize the state entropy $H(\rm{s})$ as previous data- and competence-based IM methods do. Instead, we propose to maximize a newly introduced lower bound of the conditional state entropy $H(\rm{s}|\rm{z})$, and a novel lower bound of the Mutual Information (MI) $I(\rm{s};\rm{z})$. We argue that this approach is novel and could efficiently enhance state coverage $H(\rm{s})=H(\rm{s}|\rm{z})+I(\rm{s};\rm{z})$ and promote dynamic and diverse skill discovery. Empirically, CIM for RFPT improves the performance across all standard RFPT evaluation metrics, including skill diversity (as shown in Fig. 1), state coverage (as shown in Table 2), pre-training sample efficiency (90x faster than leading methods LSD and CSD), and fine-tuning efficiency (on par with top-tier methods such as CIC, BeCL and MOSS as shown in Table 4).
>
> **Q2. I’m not sure the knowledge-based and data-based classification of methods in the introduction is entirely accurate. For example, the objective given in Zhang et al. 2021 can recover maximum entropy exploration technique in special case.**
>
> A2: The classification of methods into knowledge-based and data-based, as introduced, is widely recognized in previous works like APT (Liu & Abbeel 2021b), APS (Liu & Abbeel 2021a), URLB (Laskin et al. 2021)), MADE (Zhang et al. 2021), CIC (Laskin et al. 2023), and MOSS (Zhao et al. 2022). Regarding the paper MADE (Zhang et al. 2021) the reviewer mentioned, Zhang et al. proposed two regularizers. One is the general regularizer $R(d_\pi)$, which is equivalent to our intrinsic objective $J_i(d_\pi)$. The other is the specific MADE regularizer $E_{\rm{s}} [ (\rho_\pi^{-1}(s)d_\pi^{-1}(s))^{1/2} ]$, which belongs to knowledge-based IM methods. Thus, when we refer to the knowledge-based IM method MADE, we mean this specific MADE regularizer. Zhang et al.'s regularizer perspective aligns with our intrinsic objective perspective. Actually, all exploration techniques can be recovered with a specific intrinsic objective $J_i(d_\pi)$, as shown in the second column of Table 1. For instance, maximum entropy exploration like MaxEnt and APT can be recovered by setting $J_i(d_\pi)=H(s)=E_{\rm{s}}[-\log d_\pi(s)]$.
>
> **Q3. $\tau_k$ is more like a hyperparameter to control the trade-off between exploration and exploitation and is not really a temperature parameter—those are usually used in the form $\exp{\tau x}$ such as in softmax outputs or Boltzmann exploration.**
>
> A3: We agree with the reviewer that $\tau_k$ can also be regarded as a hyperparameter. We refer to $\tau_k$ as the temperature to maintain consistency with previous works like MADE (Zhang et al. 2021). In MADE, Zhang et al. proposed the regularized objective $J_e(d_\pi) + \tau_k R(d_\pi)$. Here, the former represents the extrinsic objective, and the latter formulates the intrinsic objective. We unify such a regularized objective in Intrinsic Motivation Reinforcement Learning (IMRL). Specifically, it is equivalent to the objective of Extrinsic Intrinsic Motivation (EIM) tasks $J_e(d_\pi) + \tau_k J_i(d_\pi)$. Therefore, we name $\tau_k$ as a temperature parameter, which determines the strength of exploration. If it would be clearer, we could revise "tempareture parameter" as "hyperparameter" as the reviewer suggests.

---

### Author Response · Authors · 2023-11-14
**General Response**

We thank the reviewers for all the detailed comments and insightful suggestions. We have highlighted the changes in blue in the revised version of our paper. Here, we provide an overview of our modifications.

(1) In Section 1, we add the discussion on the importance of the latent skill in RFPT tasks and the implications of the bias in EIM tasks.
(2) In Section 2, we add an illustration of the notations of the state entropy $H(\rm{s})$ and the conditional state entropy $H(\rm{s}|\rm{z})$ in the IM domain.
(3) In the caption of Table 1, we add the definitions of $s_T,s',z$.
(4) In Section 3.1.1, we add a clarification as to why $J_i^\text{CIM} = H(\rm{s}|\rm{z})$ depends on $\phi$.
(5) In the last line of Section 3.1.1, we add the definition for $N$.
(6) In Section 3.1.2, we update Theorem 2 to Lemma 2 and formalize the argument "the lower bound of the conditional state entropy is tight when $\phi(s)$ and $z$ is well aligned" into Theorem 3.
(7) In Section 3.1.2, we add the definition of $\rm{k}$ in the sentence "the intrinsic reward of CIM for RFPT is then...".
(8) In the sub-caption of Fig. 3(a), we include the task name for the visualization.
(9) In Appendix D, we add a proof for Theorem 3.
(10) In Appendix E, we add a Venn Diagram to clarify the relationships of all Macros involved in IMRL.
(11) In Appendix F, we add extra experiments for the evaluation of our adaptive temperature scheduler CIM for EIM.
(12) We replace all abbreviations of previous approaches with in-text citations to improve the readability of our paper.

---

### Author Response · Authors · 2023-11-22
**General Response**

We appreciate the reviewers' timely and constructive feedback on our responses. We have updated our paper based on the new comments and highlighted the changes in blue. Especially, we have made further modifications in our latest revision as follows:

(13) In Section 3.1.1 , we have adopted the more friendly notation $H^\pi(\phi(\rm{s})|\rm{z})$ in Eqn. (15) for the conditional state entropy.
(14) In Section 3.1.2, we have changed Lemma to Proposition.
(15) We have changed the $\rm{k}$ in the $\rm{k}$-nearest neighbor algorithm to $\xi$ , i.e., the $\xi$-th nearest neighbor.
(16) In Section 3.2, we have added the SGD update Equation to explicitly show the relationship between $\lambda_k$ and $R_k$.

---

### Meta-Review · Area_Chair_yT1x · 2023-12-06

**Metareview:**

The authors propose a new regularizer for mutual information-based unsupervised skill discovery. They also propose to adjust the scale of the exploration bonus based on constrained optimization in scenarios where it is traded off with extrinsic reward. The qualitative results on the Ant domain look very promising and the paper is well-situated in the previous literature. However, the quantitative performance of the method is not thoroughly evaluated outside of URLB, where the gains are marginal, and moreover the presentation of the paper is confusing. Although Appendix F.2 references Table 7 mentioning task improvements, those results don't seem to be included in the table or elsewhere in the paper. Evaluation against published methods on harder tasks would give clarity on the quantitative performance. Introducing clear structure, presenting in the main text only what is necessary for the main message, and avoiding non-standard abbreviations, and proof-reading for grammar and typos throughout the paper would improve the presentation. Although minor, the trajectory graphs should be rendered using fewer trajectories or included as bitmap images to prevent the PDF from lagging. The authors are encouraged to improve the submission of this promising algorithm and resubmit in the future.

**Justification For Why Not Higher Score:**

No convincing evaluation of task performance, presentation and writing needs improvement

**Justification For Why Not Lower Score:**

N/A

---

### Decision · Program_Chairs · 2024-01-16

Reject